# The homophilic receptor PTPRK selectively dephosphorylates multiple junctional regulators to promote cell–cell adhesion

Gareth W Fearnley[1], Katherine A Young[1], James R Edgar[1,2], Robin Antrobus[1], Iain M Hay[1], Wei-Ching Liang[3], Nadia Martinez-Martin[4], WeiYu Lin[3], Janet E Deane[1], Hayley J Sharpe[1]*

[1]Cambridge Institute for Medical Research, University of Cambridge, Cambridge, United Kingdom; [2]Department of Pathology, University of Cambridge, Cambridge, United Kingdom; [3]Antibody Engineering Department, Genentech, South San Francisco, United States; [4]Microchemistry, Proteomics and Lipidomics Department, Genentech, South San Francisco, United States

**Abstract** Cell-cell communication in multicellular organisms depends on the dynamic and reversible phosphorylation of protein tyrosine residues. The receptor-linked protein tyrosine phosphatases (RPTPs) receive cues from the extracellular environment and are well placed to influence cell signaling. However, the direct events downstream of these receptors have been challenging to resolve. We report here that the homophilic receptor PTPRK is stabilized at cell-cell contacts in epithelial cells. By combining interaction studies, quantitative tyrosine phosphoproteomics, proximity labeling and dephosphorylation assays we identify high confidence PTPRK substrates. PTPRK directly and selectively dephosphorylates at least five substrates, including Afadin, PARD3 and δ-catenin family members, which are all important cell-cell adhesion regulators. In line with this, loss of PTPRK phosphatase activity leads to disrupted cell junctions and increased invasive characteristics. Thus, identifying PTPRK substrates provides insight into its downstream signaling and a potential molecular explanation for its proposed tumor suppressor function.

DOI: https://doi.org/10.7554/eLife.44597.001

*For correspondence:
hjs49@cam.ac.uk

## Introduction

Multicellular organisms have evolved elaborate mechanisms of intercellular communication in order to organize cells into functioning tissues. The phosphorylation of protein tyrosine residues is an essential feature of cell-cell communication and effectively coordinates diverse cell behaviors such as cell adhesion and motility in response to external stimuli. Kinases and phosphatases dynamically regulate phosphotyrosine levels, such that cells are primed to acutely respond to developmental cues or changes to their local environment. In particular, enzyme-linked cell surface receptors transduce external signals to the cell interior. For example, receptor tyrosine kinases (RTKs) dimerize upon ligand binding leading to trans-autophosphorylation, which recruits phosphotyrosine-binding proteins that propagate a variety of signaling cascades (*Hunter, 2009*). Protein tyrosine phosphatases (PTPs) are often thought to function in terminating or thresholding such signals (*Agazie and Hayman, 2003*). However, it is increasingly apparent that phosphatases themselves can propagate signals in response to growth factors; with the best example being PTPN11/SHP2; a key therapeutic target in cancer (*Brown and Cooper, 1996*). Moreover, many human PTPs are receptor-linked

suggesting they can also receive input from the extracellular environment. The effects of protein phosphorylation are site-specific, for example, phosphorylation of Src Tyr419 upregulates kinase activity but phosphorylation of Tyr530 reduces it (*Mohebiany et al., 2013*). Thus, both kinases and phosphatases can modulate signaling cascades to affect cell behaviors. Despite this, the roles and substrates of the classical PTP family remain comparatively understudied.

The receptor type PTPs (RPTPs) are type one transmembrane proteins subdivided according to their extracellular domain (ECD) features. Like RTKs, RPTPs link extracellular sensing to intracellular catalysis. The regulatory mechanisms for most of the 21 RPTPs encoded by the human genome are poorly characterized (*Tonks, 2006*); however, it is known that the R2B RPTP subfamily form homophilic interactions and have been proposed to respond to cell-cell contact (*Aricescu et al., 2007*). There are four human R2B receptors: PTPRK, PTPRM, PTPRT and PTPRU, which share a common domain architecture of one MAM (meprin/A5/μ), one immunoglobulin (Ig)-like and four fibronectin (FN) domains combined with an uncharacterized juxtamembrane domain and tandem intracellular phosphatase domains; the first active (D1) and the second inactive (D2) (*Figure 1A*). Structural and biophysical studies suggest the PTPRM extracellular domain forms a rigid, pH-dependent, homophilic interaction in trans through the MAM-and Ig domains of one molecule and the FN1 and FN2 domains of another molecule, with the possibility of further *cis* interactions (*Aricescu et al., 2007*). Several cell adhesion proteins, such as cadherins and catenins, are proposed substrates for PTPRM (*Craig and Brady-Kalnay, 2015*). Its paralog PTPRK was identified as a candidate driver gene in mouse intestinal tumorigenesis by insertional mutagenesis (*March et al., 2011*; *Starr et al., 2009*) and was more recently identified as a gene fusion partner with the oncogene *RSPO3* in a subset of human colorectal cancers (*Seshagiri et al., 2012*). Furthermore, single nucleotide polymorphisms (SNPs) within the *PTPRK* genic region are associated with inflammatory bowel diseases (IBDs) and type I diabetes age of onset (*Inshaw et al., 2018*; *Trynka et al., 2011*). PTPRK is regulated by a proteolytic cascade involving furin, ADAM10 and γ-secretase (*Anders et al., 2006*) and might function to dephosphorylate proteins such as EGFR (*Xu et al., 2005*) or STAT3 (*Chen et al., 2015*). *PTPRK* mRNA is broadly expressed, except in immune cells, skeletal muscle and testes (*Figure 1—figure supplement 1A*), and is upregulated by transforming growth factor β (TGFβ) signaling (*Wang et al., 2005*). Despite its importance in disease and signaling, the events downstream of PTPRK are not well established.

Phosphatases present unique experimental challenges. For example, their signal, removal of phosphate, is inherently negative and means it is critical that they are studied in an appropriate context (*Fahs et al., 2016*). Given their homophilic interactions and subcellular localization, it is highly likely that the R2B family function at cell-cell contacts. We therefore reasoned that the appropriate context to assess their function would be in confluent, contact-inhibited epithelial monolayers. By combining proteomics approaches with in vitro and cell-based dephosphorylation assays we find that PTPRK displays striking substrate selectivity. In addition, there are distinct requirements for the two PTPRK intracellular phosphatase domains for substrate recognition. Multiple lines of evidence converge on five high confidence substrates: Afadin (AF6), PARD3 (Par3), p120$^{Cat}$ (p120-Catenin; CTNND1), PKP3 and PKP4 (p0071), which are known regulators of junctional organization. Indeed, PTPRK loss perturbs epithelial junction integrity and promotes invasive behaviors in spheroid cultures, consistent with its putative tumor suppressor role.

## Results

### PTPRK localizes to cell-cell contacts in epithelial cells

In order to detect endogenous PTPRK by immunoblot and immunofluorescence, we generated and characterized monoclonal antibodies against the purified PTPRK extracellular domain (ECD) (*Figure 1—figure supplement 1B–D*). Using one of our antibodies for structured illumination microscopy, we found PTPRK localized to puncta at basal cell-cell contacts that partially overlap with the adherens junction (AJ) protein E-Cadherin in MCF10A epithelial cells (*Figure 1B*). Homophilic interactions of the R2B receptor family have been demonstrated using suspension cell or bead aggregation assays (*Brady-Kalnay et al., 1993*; *Gebbink et al., 1993*; *Sap et al., 1994*; *Zondag et al., 1995*), and PTPRM-based structural and biophysical studies (*Aricescu et al., 2006*; *Aricescu et al., 2007*). To investigate homophilic PTPRK interactions in cells, we generated CRISPR/Cas9 PTPRK

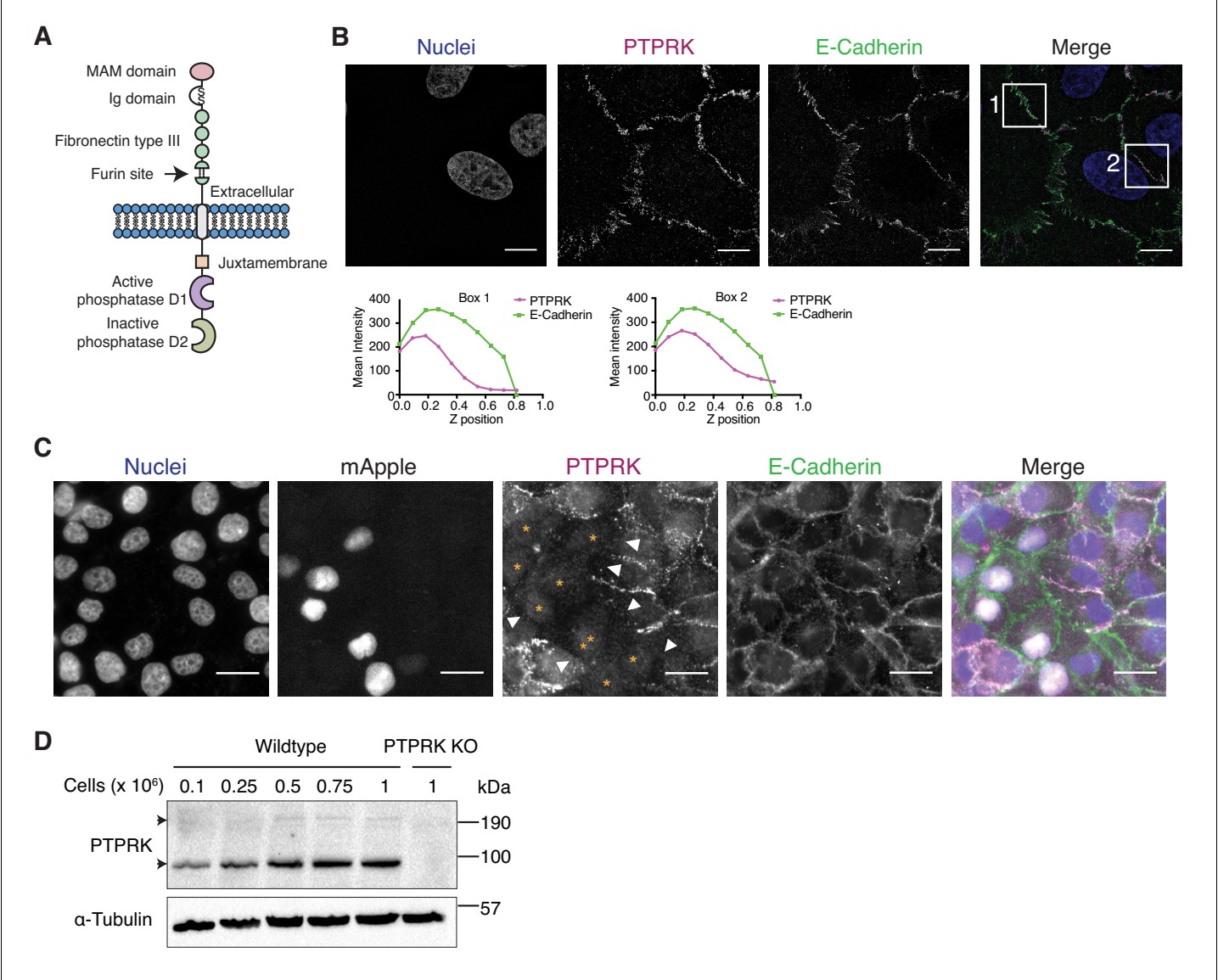

**Figure 1.** The homophilic receptor PTPRK is stabilized by cell-cell contact. (**A**) Schematic of full length PTPRK. The extracellular MAM, Ig and fibronectin domains mediate homophilic interactions. The intracellular domain comprises a juxtamembrane domain and two PTP domains; one active (D1) and one inactive (D2). (**B**) Structured illumination microscopy images of MCF10As immunostained for PTPRK (F4 clone; magenta) and E-Cadherin (green). Graphs indicate fluorescence intensity through the Z-axis in indicated boxed regions. Scale bars = 10 µm. (**C**) Fluorescence microscopy images from co-cultures of wildtype and nuclear mApple-expressing PTPRK knockout MCF10As that were immunostained for PTPRK (magenta) and E-Cadherin (green). Nuclei were stained with Hoechst (blue). mApple positive PTPRK KO cells are indicated by orange asterisks. Cell junctions where PTPRK is absent are highlighted by white arrows. Scale bars = 20 µm. (**D**) MCF10As were plated at indicated densities and analyzed by immunoblot after 3 days in culture. Arrows indicate full length (top) and furin-cleaved PTPRK (bottom). See also *Figure 1—figure supplement 1*.

DOI: https://doi.org/10.7554/eLife.44597.002

The following figure supplement is available for figure 1:

**Figure supplement 1.** Generation and validation of PTPRK antibodies and interaction screen.

DOI: https://doi.org/10.7554/eLife.44597.003

knockout (KO) MCF10A cells (*Figure 1—figure supplement 1E*), stably expressing nuclear mApple, and co-cultured them with unlabeled wildtype cells. By immunostaining, PTPRK is strikingly absent from cell-cell contacts between wildtype and adjacent KO cells, despite expression of other R2B receptors (*Figure 1C* and *Figure 1—figure supplement 1F*). Consistently, PTPRK protein levels increase with increasing cell density (*Figure 1D*). Finally, screening recombinant PTPRK ECD against

a secreted protein microarray did not identify any additional ligands (*Figure 1—figure supplement 1G*). Thus, in combination, our data indicates homophilic trans-interactions stabilize PTPRK at cell-cell contacts in epithelial cells.

## The PTPRK interactome reveals associations with cell adhesion regulators

To understand the function of PTPRK at cell-cell contacts we aimed to identify its direct substrates. Previous studies have described PTP substrate-trapping mutations, which correspond to D1057A and C1089S for the longest isoform of human PTPRK (*Flint et al., 1997*). We purified bacterially-expressed, biotinylated PTPRK wildtype and substrate-trapping intracellular domains (ICDs), as well as the pseudophosphatase D2 domain, and coupled them to streptavidin beads (*Figure 2—figure supplement 1A and B*) for affinity purification followed by mass spectrometry (AP-MS). We confirmed that the wildtype ICD could potently dephosphorylate tyrosine phosphorylated peptides, whereas the substrate traps and D2 domain were inactive, even at high concentrations (*Figure 2—figure supplement 1C*). To generate cell lysates enriched with tyrosine phosphorylated proteins, confluent MCF10A cells were treated with pervanadate, an irreversible PTP inhibitor (*Figure 2A*; *Huyer et al., 1997*). Excess vanadate was chelated with EDTA and endogenous PTP active site cysteine residues were alkylated with iodoacetamide, which was quenched by DTT, as previously described (*Blanchetot et al., 2005*). We confirmed that the substrate-trapping mutants bound tyrosine phosphorylated proteins (*Figure 2—figure supplement 1D*). Next, proteins bound to PTPRK domains after pull downs were trypsinized and identified by mass spectrometry.

Sixty-four proteins were >2 fold enriched (p<0.05; n = 4) on the wildtype PTPRK-ICD (*Figure 2B* and *Figure 2—source data 1* and *2*). We also screened for interactors using pervanadate-treated Hs27 Human fibroblast lysates (n = 3); another cell line that undergoes contact inhibition of proliferation (*Figure 2—figure supplement 2A–C*). We found that only 21% of the PTPRK-ICD interactome overlapped between MCF10A and Hs27 cells (*Figure 2—figure supplement 2D*), which might reflect differences in protein expression or phosphorylation between the cell lines. The first substrate trap (D1057A) enriched the serine/threonine kinase MAP4K4 and RAPGEF6, which were both recently linked to Hippo signaling (*Figure 2C* and *Figure 2—figure supplement 2B*; *Meng et al., 2018*). FMRP and the cell junction associated proteins PARD3, Afadin (AF-6/MLLT4) and PLEKHA6 were enriched on the second substrate trap (C1089S; *Figure 2D* and *Figure 2—figure supplement 2C*). Gene ontology (GO) term analysis for the PTPRK interactome highlights the enrichment of cell junction proteins across all domains (*Figure 2E*).

We used pull downs followed by immunoblotting to confirm interactions with previously reported PTPRK interactors (MINK1, PKP4, DLG5 and PTPN14 [*St-Denis et al., 2016*]) as well as proteins bound to the PTPRK substrate traps in this study, including FMRP-interacting NUFIP2. RAPGEF6, MAP4K4 and PARD3 were reproducibly enriched on substrate traps (*Figure 2F and G*). We did not observe interactions with previously reported R2B receptor substrates including E-Cadherin, β-Catenin, STAT3, EGFR (pY1068) and Paxillin (DEPOD database; *Duan et al., 2015*) besides p120$^{Cat}$ (*Zondag et al., 2000*), which was enriched on the C1089S trap along with PKP4 and NUFIP2 (*Figure 2G*). The principle of substrate trapping necessitates a direct interaction mediated by phosphotyrosine (*Flint et al., 1997*). We tested whether trapped proteins could be competed off PTPRK-D1057A using the phosphate mimetic orthovanadate. The MAP4K4 interaction with PTPRK D1057A ICD from pervanadate lysates was competed using orthovanadate, consistent with phosphotyrosine-mediated trapping (*Figure 2H* and *Figure 2—figure supplement 2E*). In contrast, Afadin was not depleted by orthovanadate treatment. Furthermore, PARD3, PKP4 and p120$^{Cat}$ bind the C1089S ICD less effectively in the absence of phosphorylation, also supporting the efficacy of the trapping approach (*Figure 2I*). However, we noted that all substrate-trapped proteins could still interact with PTPRK domains in phosphatase-treated lysates (*Figure 2H and I*) and most interactors can bind to the enzymatically active WT ICD (*Figure 2F and G*), indicating phosphorylation-independent PTPRK interactions. Furthermore, the PTPRK-D2 domain alone was sufficient to pull down approximately a third of PTPRK ICD interactors (*Figure 2—figure supplement 3A and B*; *Figure 2—source data 2*). Although substrate trapping was effective, our data indicate that because many proteins can bind PTPRK independently of phosphorylation or to its D2 pseudophosphatase domain (*Figure 2F and G*), trapping approaches alone could miss potential substrates.

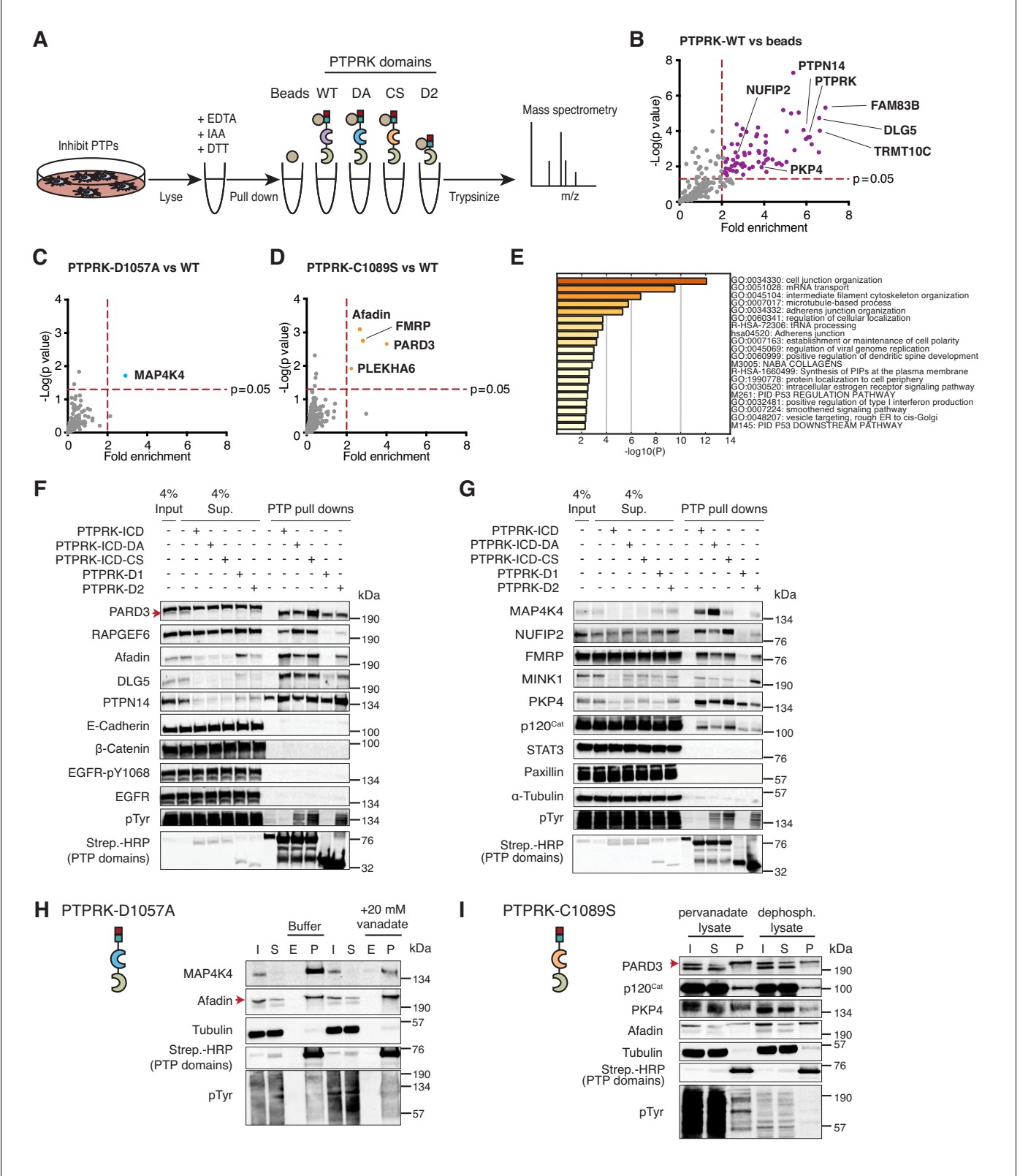

**Figure 2.** The interactome of the homophilic adhesion receptor PTPRK. (A) Experimental schematic of PTPRK interactome and substrate trapping studies. DA = D1057A, CS = C1089S. (B–D) Statistically enriched (p<0.05, n = 4) proteins after pull downs from pervanadate treated MCF10A lysates are displayed on volcano plots comparing PTPRK-ICD to beads control (B), PTPRK-ICD-DA to PTPRK-ICD (C) and PTPRK-ICD-CS to PTPRK-ICD (D). (E) GO term analysis of proteins statistically enriched (p<0.05) on PTPRK-ICD domains using Metascape. (F–G) Selected PTPRK interactors identified by mass

*Figure 2 continued on next page*

*Figure 2 continued*
spectrometry were validated by immunoblot analysis. Input and supernatants reveal the extent of protein depletion by recombinant proteins. Arrow indicates relevant band. See also *Figure 2—figure supplements 1*, *2* and *3*. (H) Confluent, pervanadate-treated MCF10A lysates were used for pull downs with PTPRK D1057A ICD. Where indicated, pull downs were incubated with and without 20 mM vanadate for 30 min. 4% inputs (I), 4% supernatants (S), 4% eluates (E; following vanadate treatment) and pull downs (P) were subjected to immunoblot analysis. (I) Confluent, pervanadate-treated MCF10A lysates were treated with or without CIP to remove protein phosphorylation and were used for pull downs with PTPRK C1089S ICD. 4% inputs (I), 4% supernatants (S) and pull downs (P) were subjected to immunoblot analysis.
DOI: https://doi.org/10.7554/eLife.44597.004

The following source data and figure supplements are available for figure 2:

**Source data 1.** Raw and processed PTPRK interactome proteomic data.
DOI: https://doi.org/10.7554/eLife.44597.008
**Source data 2.** PTPRK domain-interaction summary.
DOI: https://doi.org/10.7554/eLife.44597.009
**Figure supplement 1.** Purification of biotinylated recombinant PTPRK domains.
DOI: https://doi.org/10.7554/eLife.44597.005
**Figure supplement 2.** PTPRK interactome from Hs27 cell lysates and vanadate competition.
DOI: https://doi.org/10.7554/eLife.44597.006
**Figure supplement 3.** PTPRK-D2 interactome and PTPRM pull downs.
DOI: https://doi.org/10.7554/eLife.44597.007

To investigate interaction specificity further, we purified the ICD of the paralogous receptor PTPRM, which is 75% identical to PTPRK at the amino acid level (*Figure 2—figure supplement 3C*). Afadin, RAPGEF6 and NUFIP2 interact specifically with PTPRK, indicated by their biased depletion from supernatants. Interestingly, we found several that bound both PTPRK and PTPRM ICDs such as PARD3 and PKP4. Although MAP4K4, MINK1, PTPN14, DLG5 and p120^Cat are depleted by both ICDs, they appear to have a higher affinity for PTPRK in pull downs (*Figure 2—figure supplement 3D and E*). Overall, the PTPRK interactome is enriched with cell junction-related proteins and shows partial overlap with PTPRM. Together, these data suggest that PTPRK and PTPRM have both unique and redundant roles at cell junctions.

## The PTPRK dependent tyrosine phosphoproteome

Next, we reasoned that PTPRK deletion should result in the hyperphosphorylation of its substrates. To investigate this, we used quantitative tyrosine phosphoproteomics to compare wildtype and PTPRK KO MCF10A cells. We investigated the tyrosine phosphoproteome of confluent cells 24 hr post media change in order to observe residual phosphorylation, initially induced by EGF and/or serum growth factors. To this end, tyrosine phosphorylated peptides were enriched from trypsinized SILAC (stable isotopomeric versions of amino acids)-labeled wildtype and PTPRK KO MCF10A lysates using anti-pTyr Abs and biotin-tagged phosphotyrosine 'superbinder' mutant Src Homology 2 (SH2) domains (*Tong et al., 2017*) (*Figure 3A* and *Figure 3—figure supplement 1A*). We identified 282 quantifiable phosphotyrosine sites on 185 proteins (*Figure 3—source data 1*) from three experiments. Interestingly, 15 phosphosites were statistically upregulated in PTPRK KO cells compared to wildtype in at least two experiments, but only one site, in PAG1, was down regulated (*Figure 3B*). Strikingly, Afadin, PARD3 and PLEKHA6, which were all 'substrate-trapped' by PTPRK-C1089S, were amongst the proteins possessing enriched phosphosites in PTPRK KO cells (*Figure 3B*). Moreover, we identified upregulated phosphorylation in at least one experiment for p120^Cat, PKP2, PKP3 and PKP4, which are δ-catenin family proteins and interact with the PTPRK ICD (*Figure 3B* and *Figure 3—figure supplement 1B*). Sites on KIAA1217 and Girdin were also upregulated in PTPRK KO cells, and analysis of our raw interaction data (*Figure 2—source data 1*) showed peptides for each protein were present in PTPRK pull downs, suggesting they are also potential substrates. Unfortunately, antibodies were not available to study them further. Critically, our total proteome analysis showed that the observed phosphosite levels on all proteins were not due to differences in protein amounts, except for PLEKHA6, which was not quantified (*Figure 3C* and *Figure 3—source data 1* and *2*). Beyond PTPRK-interacting proteins we found upregulated phosphosites on several other cell-cell adhesion regulators as well as ST5, ARHGAP5 and the receptor DCBLD2 (*Figure 3B and D*). Interestingly, DCBLD2 was previously identified as a PTPRK-interacting

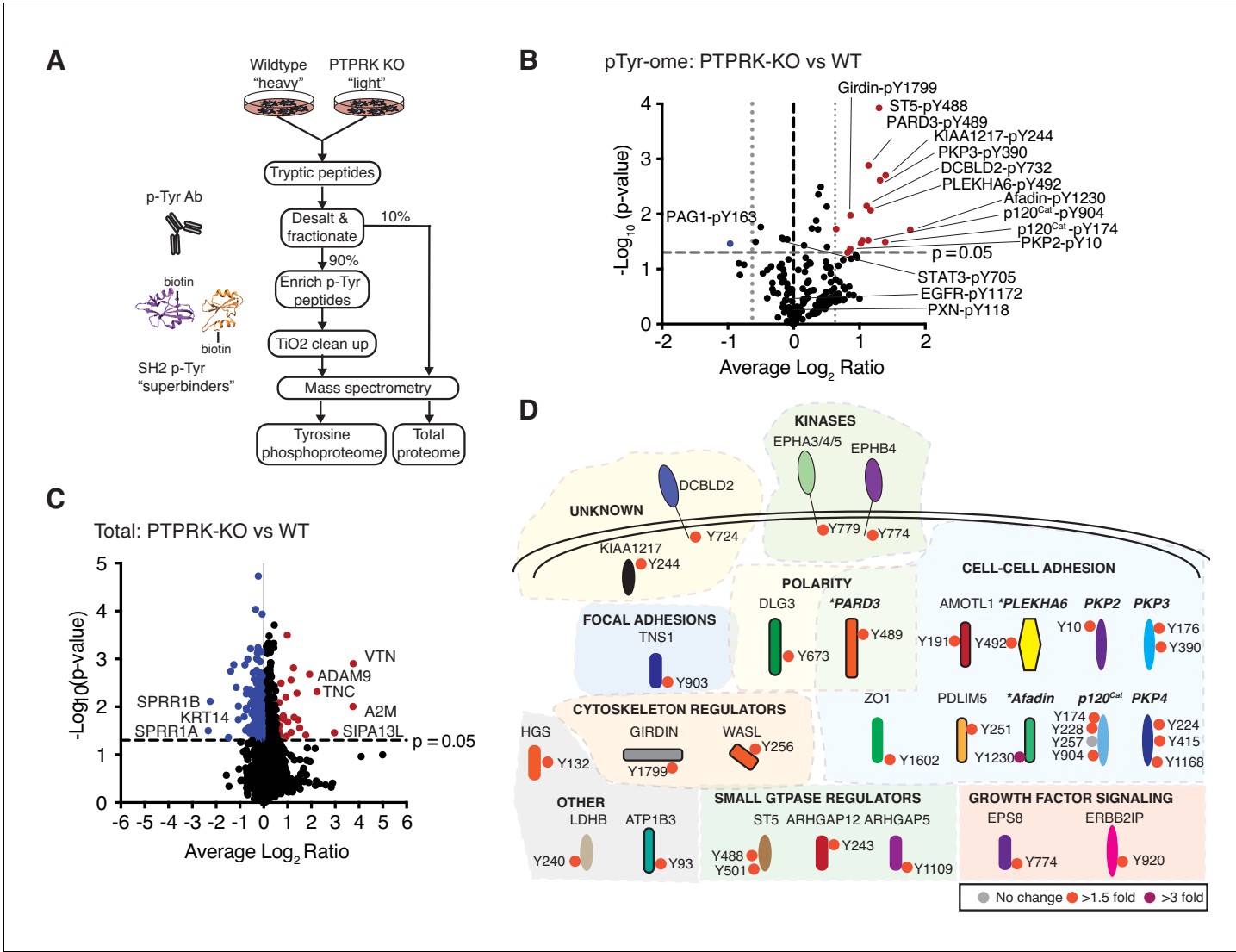

**Figure 3.** The PTPRK dependent tyrosine phosphoproteome. (**A**) Schematic of workflow to enrich and identify phosphotyrosine peptides from SILAC-labeled wildtype and PTPRK KO MCF10As. Equal amounts of wildtype and PTPRK KO cell lysates were combined prior to trypsinization. A 10% sample was reserved for total proteome analysis. Tyrosine phosphorylated peptides were enriched using anti-phosphotyrosine antibodies and SH2 domain 'superbinders'. (**B**) Volcano plot of tyrosine phosphosites detected in PTPRK KO and wildtype MCF10As. Phosphosites > 50% enriched in (p<0.05; n = 3) in PTPRK KO cells are labeled red and those enriched in wildtype are blue. FDR = 0.01, two valid values required. (**C**) Volcano plot of protein abundance. Proteins > 50% more abundant (p<0.05; n = 3) in PTPRK KO MCF10As are shown in red, and wildtype in blue. FDR = 0.01, two valid values required. (**D**) Overview of proteins with at least one tyrosine phosphorylation site increased in PTPRK KO cells as determined by quantitative proteomics (FDR = 0.01, one valid value required). Tyrosine phosphosite change in PTPRK KO cells compared to wildtype is indicated by colored circles:>3 fold up; purple,>1.5 fold up; red,<1.5 fold up or down (no change); grey. Proteins identified as interactors by AP-MS or immunoblotting in this study are highlighted in bold and italics. *Denotes proteins enriched on substrate traps. See also *Figure 3—figure supplements 1* and *2*.

DOI: https://doi.org/10.7554/eLife.44597.010

The following source data and figure supplements are available for figure 3:

**Source data 1.** Quantitative total and tyrosine phosphoproteomics.
DOI: https://doi.org/10.7554/eLife.44597.013
**Source data 2.** Statistically upregulated proteins and phosphotyrosine sites in PTPRK KO cells following quantitative proteomics.
DOI: https://doi.org/10.7554/eLife.44597.014
**Figure supplement 1.** The PTPRK-dependent tyrosine phosphoproteome.
DOI: https://doi.org/10.7554/eLife.44597.011
**Figure supplement 2.** The PTPRK-dependent tyrosine phosphoproteome is enriched for cell junction organization proteins.
DOI: https://doi.org/10.7554/eLife.44597.012

protein in a large-scale AP-MS study (*Huttlin et al., 2017*). In contrast, specific sites on Paxillin, EGFR and STAT3 were not changed or were undetectable (*Figure 3B* and *Figure 3—source data 1*). Thus, by combining the PTPRK interactome with tyrosine phosphoproteomics we have identified eight candidate substrates (*Figure 3—figure supplement 1C*).

Using these candidate substrates, we next aimed to determine any sequence selectivity by PTPRK. Previously, *Barr et al. (2009)* tested recombinant PTPs against a panel of phosphopeptides and observed limited sequence selectivity. For example, PTPRK showed reduced activity against peptides with basic residues in the three positions N-terminal to phosphotyrosine, including an EGFR-pY1068-containing peptide. In contrast, PTPRB showed no sequence preference (*Barr et al., 2009*). To investigate whether our candidate substrates shared common features we generated a consensus sequence, which showed a slight bias against basic residues immediately adjacent to the phosphotyrosine (*Figure 3—figure supplement 1D*). This is consistent with the positively charged PTPRK active site entrance observed in its crystal structure, which may preclude binding of positively charged or basic amino acids (*Figure 3—figure supplement 1E*). We next searched the phosphosite plus database with a seven amino acid consensus sequence phosphotyrosine and cross-referenced to the PTPRK interactome (*Figure 2—source data 1*). Beyond the candidate substrates, we identified an additional 18 phosphosites matching the consensus including substrate-trapped MAP4K4 and junction-associated ABLIM3 (*Matsuda et al., 2010*). In contrast, when we scrambled the consensus sequence we found fewer PTPRK interactors were identified (*Figure 3—figure supplement 1F*). Therefore, PTP substrate consensus sequences might be useful in expanding a candidate substrate list when interactors are known, but most likely only represents a permissive sequence for dephosphorylation, rather than a strict requirement.

Based on our data and these analyses, PKP3, MAP4K4 and ABLIM3 were also included as candidate substrates after confirming interactions with PTPRK and PTPRM domains (*Figure 3—figure supplement 2A and B*). A GO term analysis of statistically-enriched phosphosites from at least one sample (*Figure 3D* and *Figure 3—figure supplement 2C*) showed a bias towards proteins with roles in cell junction and actin cytoskeleton organization. Interestingly, several phosphosites identified here are growth factor- and, in most cases, Src kinase-dependent (*Reddy et al., 2016*). Importantly, however, the Src family kinase activating phosphotyrosine (e.g. Src-Y419) is 1.6-fold lower in PTPRK KO cells, therefore such kinase activity does not explain the observed differences (*Figure 3—source data 1*). Thus, PTPRK influences the tyrosine phosphorylation of numerous interacting proteins in cells, suggesting it has non-redundant cellular phosphatase activity.

## PTPRK interacts with candidate substrates in confluent MCF10A cells

To investigate proximity interactions of proteins identified by AP-MS and phosphoproteomics in confluent cells we used BioID (*Roux et al., 2012*). We confirmed the cell surface localization of mutant BirA and flag-tagged PTPRK-C1089S and truncated PTPRK, lacking an ICD, by immunostaining. Truncated PTPRK showed notably stronger staining at the cell surface than PTPRK-C1089S, perhaps reflecting the loss of an endocytic or degradative signal in the ICD (*Figure 4—figure supplement 1A*). Immunoblots of pulldowns from doxycycline-induced, confluent cells revealed enrichment on PTPRK-C1089S over the truncated form for the candidate substrates Afadin, PARD3, p120$^{Cat}$, PKP3, PKP4, as well as ABLIM3 and EGFR, but not E-Cadherin, Paxillin, β-Catenin, Tubulin or ZO2 (*Figure 4A and B*, and *Figure 4—figure supplement 1B*). Importantly, MINK1, PKP4, PTPN14 and DLG5 were also enriched and were previously identified by PTPRK BioID in HEK293 cells (*St-Denis et al., 2016*), lending additional support to our observations (*Figure 4C*). These data confirm that several of the interactors identified by AP-MS and phosphoproteomics experiments also interact with PTPRK in confluent MCF10A cells.

## PTPRK directly and selectively dephosphorylates polarity and junctional proteins

We next sought to determine whether PTPRK could directly dephosphorylate any of its binding partners in vitro, with a particular focus on proteins that were hyperphosphorylated in PTPRK KO cells. Phosphatases have a reputation for promiscuity therefore we included the intracellular domain of the closely related receptor PTPRM and assayed a panel of negative controls. Using an in vitro para-nitrophenylphosphate (pNPP) colorimetric dephosphorylation assay, we determined that a three-

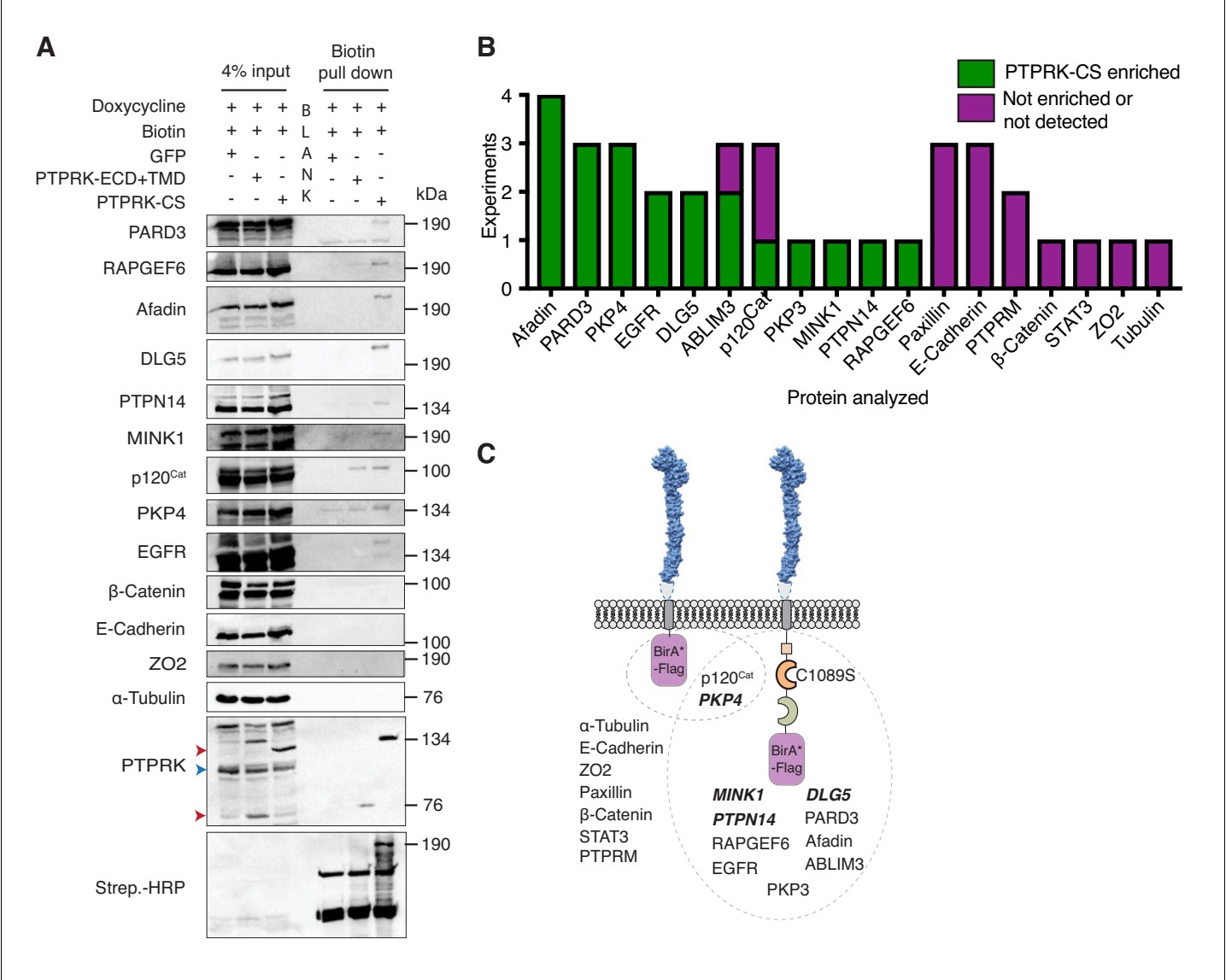

**Figure 4.** PTPRK interacts with candidate substrates in confluent MCF10A cells. (**A**) Representative immunoblot analysis of biotin pull downs from MCF10As expressing tGFP or PTPRK BioID constructs. See Materials and methods for details. Red and blue arrows indicate exogenous and endogenous PTPRK, respectively. (**B**) Quantification of BioID immunoblots. Green bars indicate the number of times a protein was enriched on PTPRK-C1089S.BirA*-Flag, compared to PTPRK.ECD +TMD.BirA*-Flag in separate experiments. Purple bars indicate the number of times a protein was not enriched or was not detected in any pull downs. n ≥ 1. (**C**) Schematic representation of PTPRK proximity-labeling by BioID. PTPRK extracellular domain homology model is based on PTPRM (PDB: 2V5Y; *Aricescu et al., 2007*). Proteins within the dotted lines were detected in pull downs from indicated BioID lysates. Proteins not detectably biotinylated are listed on the left. Proteins in bold and italics were previously identified as PTPRK interactors using BioID in HEK293 cells (*St-Denis et al., 2016*). See also *Figure 4—figure supplement 1*.

DOI: https://doi.org/10.7554/eLife.44597.015

The following figure supplement is available for figure 4:

**Figure supplement 1.** Localization of PTPRK BioID proteins.

DOI: https://doi.org/10.7554/eLife.44597.016

fold higher molar ratio of PTPRM was required to match PTPRK activity (*Figure 5—figure supplement 1A*), consistent with a previous study (*Barr et al., 2009*). Interestingly, a three-fold higher molar ratio of PTPRK-ICD was required for equivalent activity to the D1 domain, suggesting the D2 reduces D1 enzyme activity (*Figure 5—figure supplement 1A*).

To identify proteins dephosphorylated by PTPRK, pervanadate-treated MCF10A cell lysates were incubated with recombinant protein domains, followed by phosphotyrosine immunoprecipitation and immunoblotting (*Figure 5A*). We expected dephosphorylated proteins to be depleted from immunoprecipitates but present in supernatants, or, as observed for PKP4, to show a shift in molecular weight. In these assays phosphoproteins from different reactions were equally enriched by IP, as indicated by phosphotyrosine immunoblots, but the lysates incubated with active phosphatase domains had fewer phosphoproteins overall based on depletion from supernatants (Lower panel; *Figure 5B*). Consistent with our interaction data, several previously reported R2B receptor substrates including E-Cadherin, STAT3, β-Catenin, Paxillin and EGFR-pY1068 (*Duan et al., 2015*), were not dephosphorylated by either PTPRK or PTPRM under these conditions (*Figure 5B* and *Figure 5—figure supplement 1B-C*). In contrast, the PTPRK ICD, but strikingly not the PTPRK-D1 or PTPRM domains, completely dephosphorylated Afadin (*Figure 5B* and *Figure 5—figure supplement 1B*), suggesting a combined role for the D1 and D2 domains in its recognition and selective dephosphorylation. PARD3 and PKP3 were preferentially dephosphorylated by the PTPRK and PTPRM ICDs. In contrast, the PTPRK and PTPRM D1 domains alone were sufficient to dephosphorylate ABLIM3, PKP4 and p120$^{Cat}$ (*Figure 5B*). Conversely, RAPGEF6 and MINK1 were not clearly dephosphorylated by the domains under these conditions (*Figure 5—figure supplement 1B and C*). MAP4K4, FMRP and NUFIP2 were not detectably tyrosine phosphorylated in the cell lysates, precluding us from assessing dephosphorylation (*Figure 5—figure supplement 1C*). It has been suggested for the R2A RPTPs that the inactive D2 domain can inhibit the D1 domain (*Wallace et al., 1998*). However, addition of PTPRK-D2 to PTPRK-D1 did not affect its activity against, for example, p120$^{Cat}$ (*Figure 5B*). In combination with our interaction studies, these data suggest that PTPRM and PTPRK have overlapping substrate specificities for δ-catenin proteins, ABLIM3 and PARD3, and the PTPRK-ICD selectively dephosphorylates Afadin.

To further investigate the role of the PTPRK and PTPRM domains in substrate selectivity, we generated chimeric proteins consisting of combinations of the PTPRK and PTPRM D1 and D2 domains (*Figure 5C* and *Figure 5—figure supplement 1D*). In pull down assays, we found that PARD3 and p120$^{Cat}$ bound to all proteins (*Figure 5C*). Consistent with our previous findings, Afadin and NUFIP2 showed a preference for proteins with the PTPRK-D2 domain, which is particularly evident by supernatant depletion (*Figure 5C*). In contrast, RAPGEF6 bound equally to both PTPRK domains (*Figure 5C*). In dephosphorylation assays, proteins with the PTPRM D1 were used at a 3-fold higher concentration than PTPRK D1 domains to compensate for their lower activity (*Figure 5—figure supplement 1E*). Strikingly, the PTPRK-D2 domain is sufficient to recruit Afadin for dephosphorylation by PTPRM-D1 (*Figure 5D*). In contrast, p120$^{Cat}$ is dephosphorylated by all domains, based on its presence in the associated supernatants (*Figure 5D*). Consistent with our previous findings, Paxillin is not dephosphorylated by any PTPRK or PTPRM combinations (*Figure 5D*). Together, these data demonstrate that PTPRK and PTPRM can directly and selectively dephosphorylate substrates, and that the D2 pseudophosphatase domain is necessary and sufficient for recruitment of the PTPRK-specific substrate, Afadin.

A role for RPTP pseudophosphatase domains in substrate recognition has been proposed, however, the mechanism remains elusive. Structural studies on other RPTP D2 domains show a canonical PTP fold (*Barr et al., 2009*; *Nam et al., 1999*; *Nam et al., 2005*) and resemble substrate traps due to amino acid variation in key catalytic motifs. For example, the LAR (PTPRF) D2 domain could be converted to an active PTP by just two mutations (*Nam et al., 1999*). We were unable to deplete Afadin from the PTPRK D2 domain with vanadate or dephosphorylation of cell lysates, suggesting binding is not mediated by phosphotyrosine (*Figure 5—figure supplement 2A and B*). We next attempted to reactivate the PTPRK D2 domain by reintroducing canonical sequences to the WPD loop, PTP signature motif and Q loop (*Figure 5—figure supplement 2C*; *Andersen et al., 2001*). Using a pNPP assay, we found no impact of the mutations on D2 domain activity (*Figure 5—figure supplement 2D*), similar to recent failed attempts to reactivate the PTPRE D2 domain (*Lountos et al., 2018*). Importantly, the D2 domain mutations did not abrogate binding to several interactors (*Figure 5—figure supplement 2E*). The catalytic cysteine, which forms a phosphocysteine intermediate in PTP D1 domains (*Pannifer et al., 1998*), is conserved in most RPTP D2 domains (*Andersen et al., 2001*). However, the CD45 (PTPRC) D2 domain structure shows that this key cysteine is occluded when compared to that of the D1 (*Figure 5—figure supplement 2F*). We generated a homology model for the PTPRK D2 domain based on PTPRE, and found that the surface

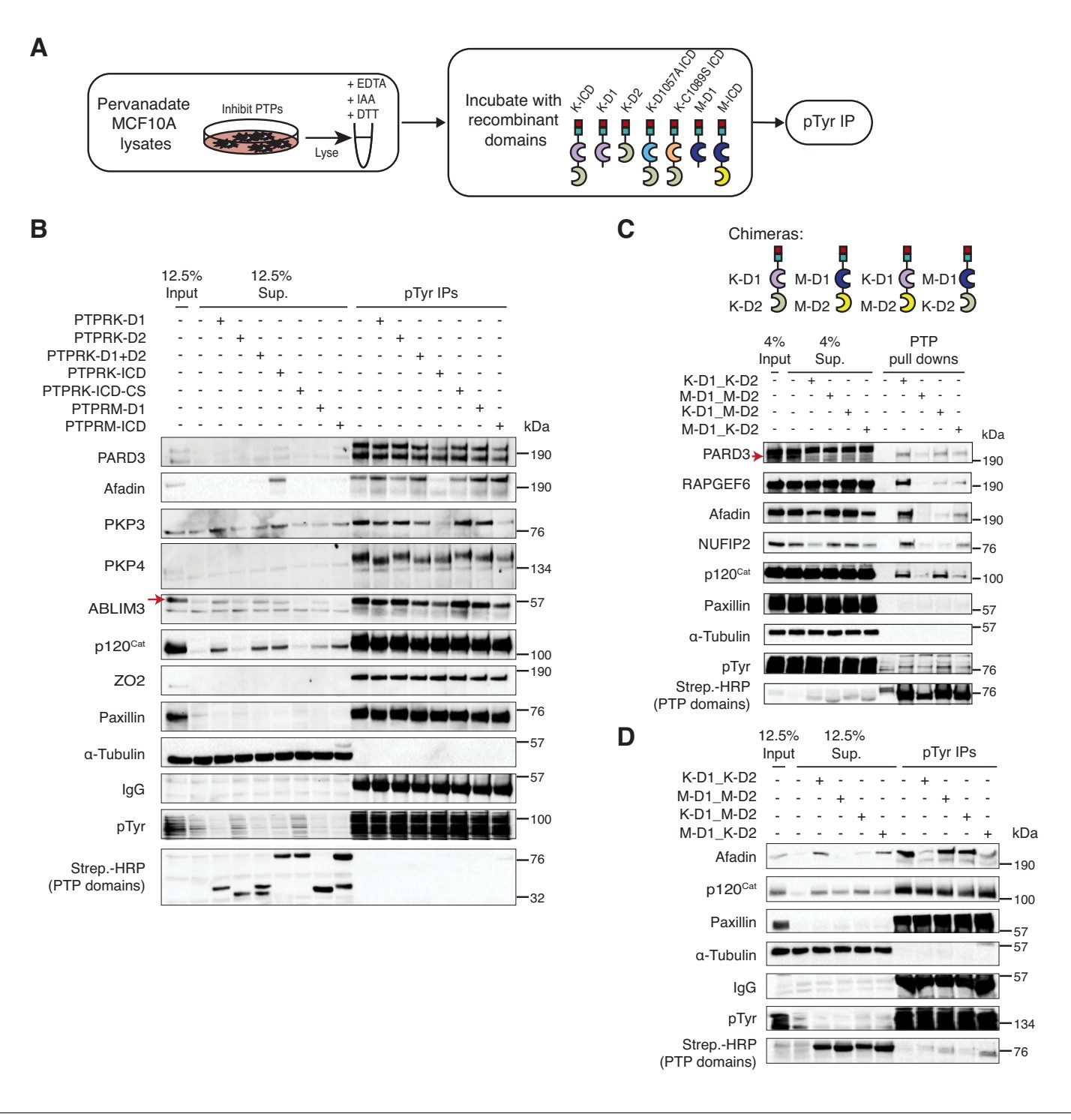

**Figure 5.** PTPRK directly and selectively dephosphorylates cell junction regulators. (**A**) Workflow of in-lysate dephosphorylation assay. Recombinant PTPRK and PTPRM domains were incubated with pervanadate-treated MCF10A lysates for 1.5 hr at 4°C, followed by immunoprecipitation of tyrosine phosphorylated proteins. (**B**) Pervanadate-treated MCF10A lysates were incubated with the indicated domains at an amount pre-determined to give equal phosphatase-activity prior to phosphotyrosine immunoprecipitation and immunoblot analysis. (**C**) Pull downs using chimeric RPTPs from confluent, pervanadate-treated MCF10A lysates were subjected to immunoblot analysis. (**D**) Pervanadate-treated MCF10A lysates were incubated with the indicated domains prior to phosphotyrosine immunoprecipitation and immunoblot analysis. See also *Figure 5—figure supplements 1* and *2*.
DOI: https://doi.org/10.7554/eLife.44597.017

The following figure supplements are available for figure 5:

*Figure 5 continued on next page*

*Figure 5 continued*

**Figure supplement 1.** In vitro dephosphorylation assays and generation of RPTP chimeras.
DOI: https://doi.org/10.7554/eLife.44597.018
**Figure supplement 2.** Analysis of PTPRK-D2 domain interactions.
DOI: https://doi.org/10.7554/eLife.44597.019

charge surrounding the putative active site significantly diverges from that of the D1 domain (*Figure 5—figure supplement 2G*). These data suggest the D2 domain substrate recognition mechanism does not require substrate phosphorylation, which is consistent with our earlier findings (*Figure 2H and I*) as well as PTPRK D2 domain structural and sequence features.

## PTPRK dephosphorylates p120$^{Cat}$-pY228 and -pY904 in MCF10A cells

The Phosphosite plus database includes 17 frequently phosphorylated Human p120$^{Cat}$ tyrosine residues that have been identified by mass spectrometry (*Hornbeck et al., 2015*). By the same criteria, the larger protein Afadin is phosphorylated on only five tyrosine residues (*Hornbeck et al., 2015*). This difference might explain why PTPRK completely dephosphorylates most of the Afadin present in lysates, but only a fraction of p120$^{Cat}$ (*Figure 5B*). Therefore, our dephosphorylation assays are likely to be quite conservative, particularly for proteins with many phosphosites. Our phosphoproteomics data revealed hyperphosphorylation of p120$^{Cat}$-Y174, -Y228, -Y865 and -Y904 in PTPRK KO cells, suggesting these could be direct targets for PTPRK. Antibodies were available to detect phosphorylated p120$^{Cat}$-Y228 and -Y904. To determine whether these sites could be directly dephosphorylated we incubated pervanadate lysates with recombinant protein domains and immunoblotted for specific phosphosites. In all cases, PTP domains did not dephosphorylate EGFR-pY1068 or Paxillin-pY118 (*Figure 6A* and *Figure 6—figure supplement 1A and B*). In contrast, PTPRK-D1 and ICD, but not the catalytically inactive PTPRK-C1089S (*Figure 6A*) or pervanadate-inhibited PTPRK ICD (*Figure 6—figure supplement 1A*), almost completely dephosphorylated both p120$^{Cat}$-pY228 and -pY904 sites. PTPRM also dephosphorylated both sites (*Figure 6—figure supplement 1B*). Whilst these p120$^{Cat}$ sites are efficiently dephosphorylated by PTPRK, only a small fraction of p120$^{Cat}$ undergoes complete dephosphorylation (*Figure 5B*). Combined with the observation that p120$^{Cat}$-pY257 levels were unchanged in PTPRK KO cells by phosphoproteomics (*Figure 3—source data 1*) our data are consistent with PTPRK site selectivity, at least for p120$^{Cat}$.

Immunoblotting confirms that p120$^{Cat}$-pY228 and -pY904 are increased on average 3–4-fold in confluent PTPRK KO cells compared to wildtype, whereas Paxillin-pY118 is unchanged, consistent with our phosphoproteomics results (n = 4; *Figure 6B and C*; *Figure 3—source data 2*). We next used the site-specific p120$^{Cat}$ phosphoantibodies to assess whether the direct dephosphorylation of putative substrates observed in vitro translated to an intact cellular context. Doxycycline-induction of PTPRK, but not the C1089S mutant, in PTPRK KO cells reproducibly and dose-dependently reduced p120$^{Cat}$-pY228 and -pY904 levels, without affecting total p120$^{Cat}$ (n = 5; *Figure 6D and E*). Conversely, reintroduction of PTPRK did not affect Paxillin-pY118. These results suggest PTPRK is an active and selective tyrosine phosphatase for p120$^{Cat}$ in confluent MCF10A cells.

## PTPRK promotes junction integrity in epithelial cells

We have demonstrated that Afadin, PARD3, PKP3, PKP4, and p120$^{Cat}$ are high confidence substrates for PTPRK, with PLEKHA6, MAP4K4, PKP2, KIAA1217, ABLIM3 and Girdin also being good candidates. Because these proteins are linked by roles in cell-cell junction organization, we sought to determine the impact of PTPRK loss on MCF10A morphology. Wildtype and PTPRK KO cells displayed signaling differences by phosphoproteomics at 24 hr after media change (*Figure 3B*). We therefore used the same conditions to investigate junctional integrity of confluent cells. To this end, wildtype and PTPRK KO cells grown on transwell filters were analyzed by electron microscopy. Wildtype cells were more closely packed and organized than PTPRK KO cells, which exhibited large gaps between cells (*Figure 7A*). Moreover, we observed a striking reduction in cell height in PTPRK KO cells (*Figure 7A* (inset) and 7B). We further investigated junctional integrity by measuring the transepithelial electrical resistance (TEER) and FITC dextran permeability of cells grown on transwell filters. PTPRK KO cells exhibited a ~ 50% reduction in TEER and a small but significant increase in FITC-

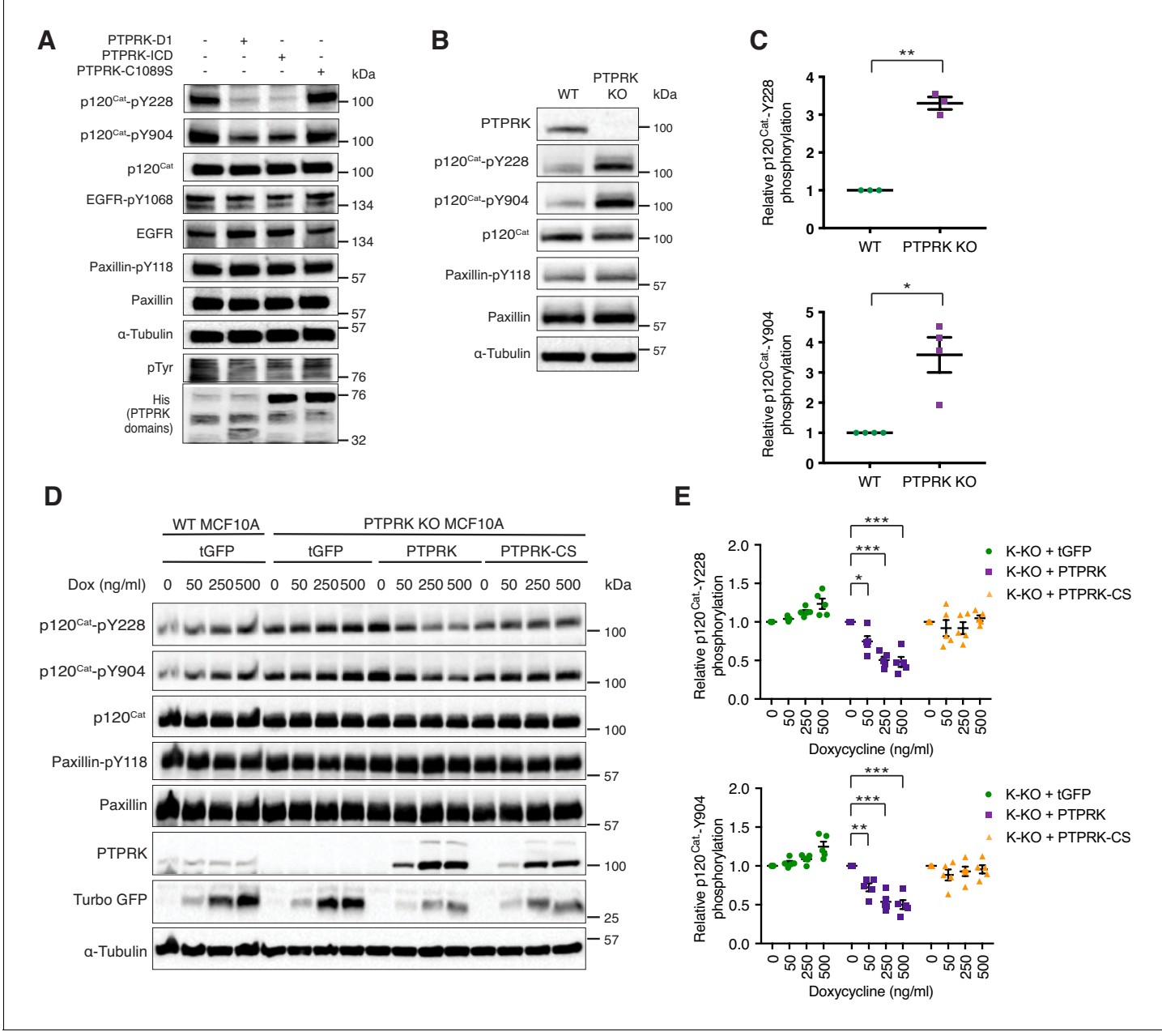

**Figure 6.** PTPRK dephosphorylates p120$^{Cat}$ Y228 and Y904 in MCF10A cells. (**A**) Pervanadate-treated MCF10A lysates were incubated with and without the indicated recombinant PTPRK-D1, PTPRK-ICD or PTPRK-C1089S-ICD for 1.5 hr at 4°C, prior to immunoblot analysis. (**B–C**) Lysates from confluent wildtype and PTPRK KO MCF10As were analyzed by immunoblot and quantified by densitometry. Error bars denote ±SEM (n ≥ 3). Unpaired, two-tailed t test: *p<0.05, **p<0.005. (**D**) Wildtype or PTPRK KO MCF10As, with stably-integrated doxycycline-inducible tGFP, PTPRK or PTPRK-C1089S, were cultured for 6 days with indicated concentrations of doxycycline then lysed and subjected to immunoblot analysis. (**E**) Densitometric quantification of p120$^{Cat}$ phosphorylation normalized against total p120$^{Cat}$. Error bars denote ±SEM (n = 5). Two-way ANOVA (Tukey's multiple comparisons test): *p<0.005**, p<0.005, ***p<0.0005. See also *Figure 6—figure supplement 1*.

DOI: https://doi.org/10.7554/eLife.44597.020

The following source data and figure supplement are available for figure 6:

**Source data 1.** Densitometric analysis of immunoblots.
DOI: https://doi.org/10.7554/eLife.44597.022
**Figure supplement 1.** PTPRK dephosphorylates p120Cat-Y228 and Y904.
DOI: https://doi.org/10.7554/eLife.44597.021

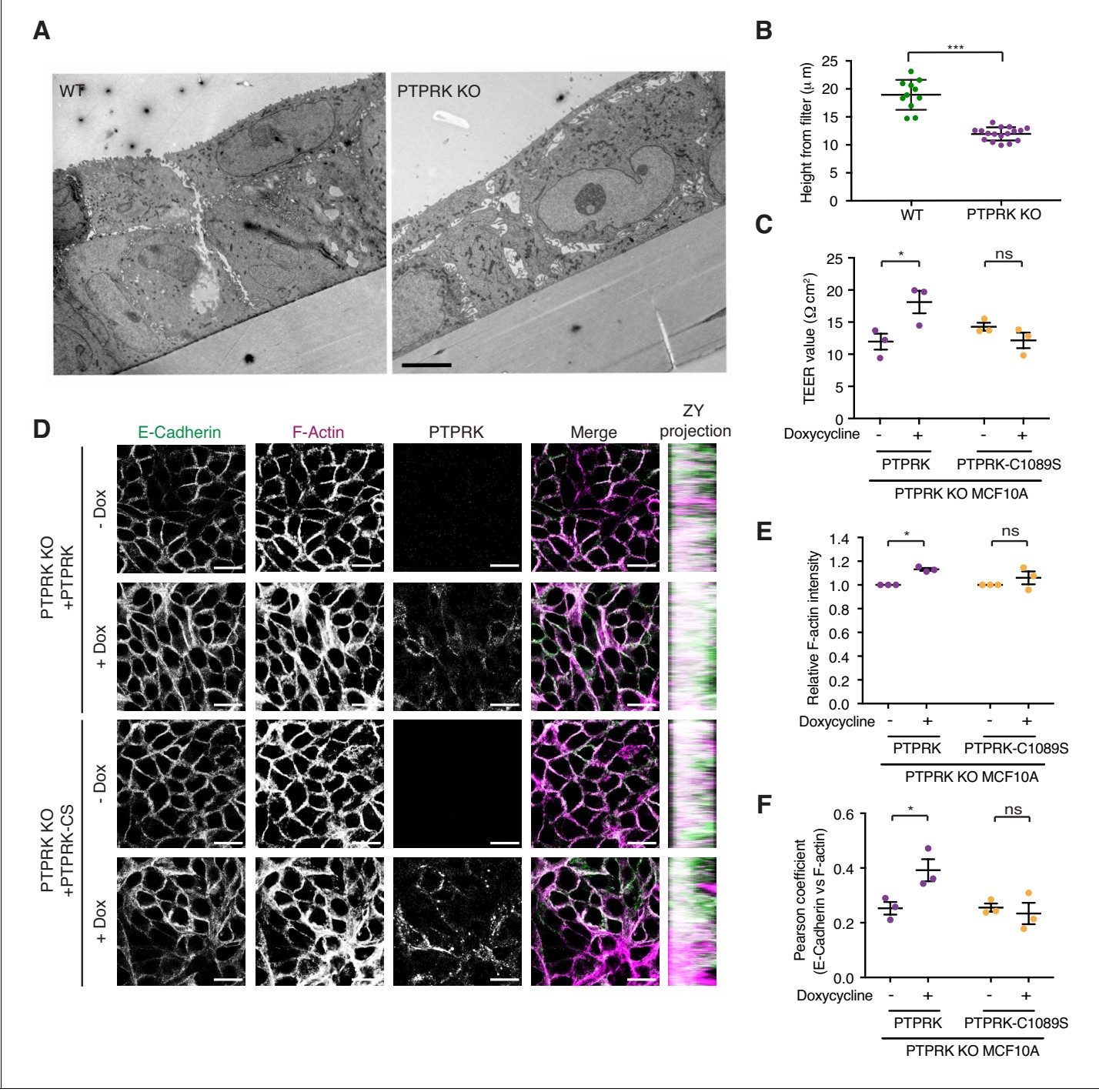

**Figure 7.** PTPRK promotes junction integrity and organization in epithelial cells. (**A**) Wildtype (Left) and PTPRK KO (Right) MCF10As were cultured on transwell filters before being fixed and prepared for conventional electron microscopy (EM). Scale bar = 5 µm. (**B**) Quantification of cell height relative to transwell filter. Three measurements per image were averaged. Each data point relates to one EM image. Error bars denote ±SEM. Unpaired, two tailed t test ***p<0.0005. (**C**) Stable PTPRK KO MCF10As were grown to confluence with or without 250 ng/ml doxycycline on 0.4 µm transwell filters prior to TEER analysis. Error bars denote ±SEM (n = 3). Two-way ANOVA (Sidak's multiple comparisons test): *p<0.05. (**D**) Confluent PTPRK KO MCF10As, with stably-integrated doxycycline-inducible PTPRK or PTPRK-C1089S, were cultured for 6 days with or without 250 ng/ml doxycycline then fixed and stained for E-Cadherin and F-actin. A representative confocal microscopy image is shown. Scale bar = 20 µm. (**E**) Quantification of relative F-actin staining intensity. 10 random fields/replicate were averaged. Error bars denote ±SEM (n ≥ 3). Two-way ANOVA (Sidak's multiple comparisons test): *p<0.05 (**F**) Quantification of colocalization (Pearson coefficient) between E-Cadherin and F-actin staining. 10 random fields/biological replicate

*Figure 7 continued on next page*

*Figure 7 continued*

were averaged. Error bars denote ±SEM (n = 3). Two-way ANOVA (Sidak's multiple comparisons test): *p<0.05. See also *Figure 7—figure supplement 1*.

DOI: https://doi.org/10.7554/eLife.44597.023

The following source data and figure supplements are available for figure 7:

**Source data 1.** Source data used in graphs.

DOI: https://doi.org/10.7554/eLife.44597.026

**Figure supplement 1.** Loss of PTPRK compromises cell junction integrity.

DOI: https://doi.org/10.7554/eLife.44597.024

**Figure supplement 2.** Both PTPRK and PTPRK-C1089S partially rescue E-Cadherin intensity.

DOI: https://doi.org/10.7554/eLife.44597.025

dextran permeability compared to wildtype cells indicating a leakier monolayer (*Figure 7—figure supplement 1A–B*). TEER measurements were partially rescued by reintroduction of PTPRK, but not a catalytically inactive mutant (*Figure 7C*). Consistent with this altered organization, immunostained PTPRK KO cells grown on coverslips displayed a ~ 20% decrease in the intensity of F-actin, the AJ protein E-Cadherin, the desmosomal protein Desmoglein 3 (DSG3) and the PTPRK substrate p120$^{Cat}$ (*Figure 7—figure supplement 1C–F*). The junctional markers also displayed reduced colocalization with F-actin (*Figure 7—figure supplement 1G–H*). However, the levels of these junctional proteins were unaffected by PTPRK loss (*Figure 6B* and *Figure 7—figure supplement 1I*). Reintroduction of PTPRK or PTPRK-C1089S was able to partially rescue E-Cadherin intensity (*Figure 7D* and *Figure 7—figure supplement 2A*), however, catalytic activity was required to rescue F-actin intensity (*Figure 7D–E*). In line with this, rescue of E-Cadherin and F-actin colocalization requires PTPRK D1 domain activity, suggesting PTPRK substrate hyperphosphorylation contributes to impaired junctional integrity.

It has previously been reported that shRNAs targeting PTPRK in MCF10A cells perturbs their morphogenesis in 3D culture (*Ramesh et al., 2015*). We find PTPRK KO cells mostly form normal acini (*Figure 8* and *Figure 8—figure supplement 1A*); however,~20% exhibited a branched or protrusive morphology after 14 days in culture (*Figure 8B*), resembling the previously described invasive behavior observed upon combined EGFR and Src overexpression in MCF10A cells (*Dimri et al., 2007*). When we collected intact spheroids for immunostaining we found normal apical polarization of the Golgi (*Figure 8C*). However, PTPRK KO spheroids were significantly larger, by diameter, than wildtype (*Figure 8D*), despite similar proliferation rates of subconfluent cells in 2D (*Figure 8—figure supplement 1B*). Overall, our results support a role for PTPRK in promoting cell-cell junctions and repressing invasive behavior, probably through recruitment and dephosphorylation of several cell junction organizers.

## Discussion

We have used unbiased approaches to identify five high confidence substrates of the cell-contact sensing receptor PTPRK, including Afadin, PARD3, p120$^{Cat}$, PKP3 and PKP4. These substrates are linked to cell-cell junction organization, which is perturbed when PTPRK is deleted. Importantly, our findings demonstrate the substrate selectivity of this receptor, which requires both its active and inactive PTP domains. We also identify PTPRK as a key mediator of adhesive signaling. Our conclusions have implications not only for understanding PTP biology and cell-cell junction phosphoregulation, but also provide molecular insight into how PTPRK might function as a tumor suppressor.

Cross-referencing the PTPRK interactome with the PTPRK-dependent tyrosine phosphoproteome enabled us to identify candidate substrates to assay for cellular interactions and direct dephosphorylation. Substrate-trapping methods in combination with mass spectrometry are commonly used to identify PTP substrates (*Blanchetot et al., 2005*). We used two mutants affecting the WPD catalytic motif (D1057A) and catalytic cysteine within the PTP signature motif (C1089S) and found enrichment of distinct proteins on each. Using both Hs27 and MCF10A lysates, MAP4K4 and RAPGEF6 were enriched on the D1057A trap. Both were partially competed from traps with the phosphate mimetic vanadate, suggesting phosphorylation-dependent interaction. However, we could not validate RAPGEF6 or MAP4K4 as PTPRK substrates by in vitro dephosphorylation or phosphoproteomics,

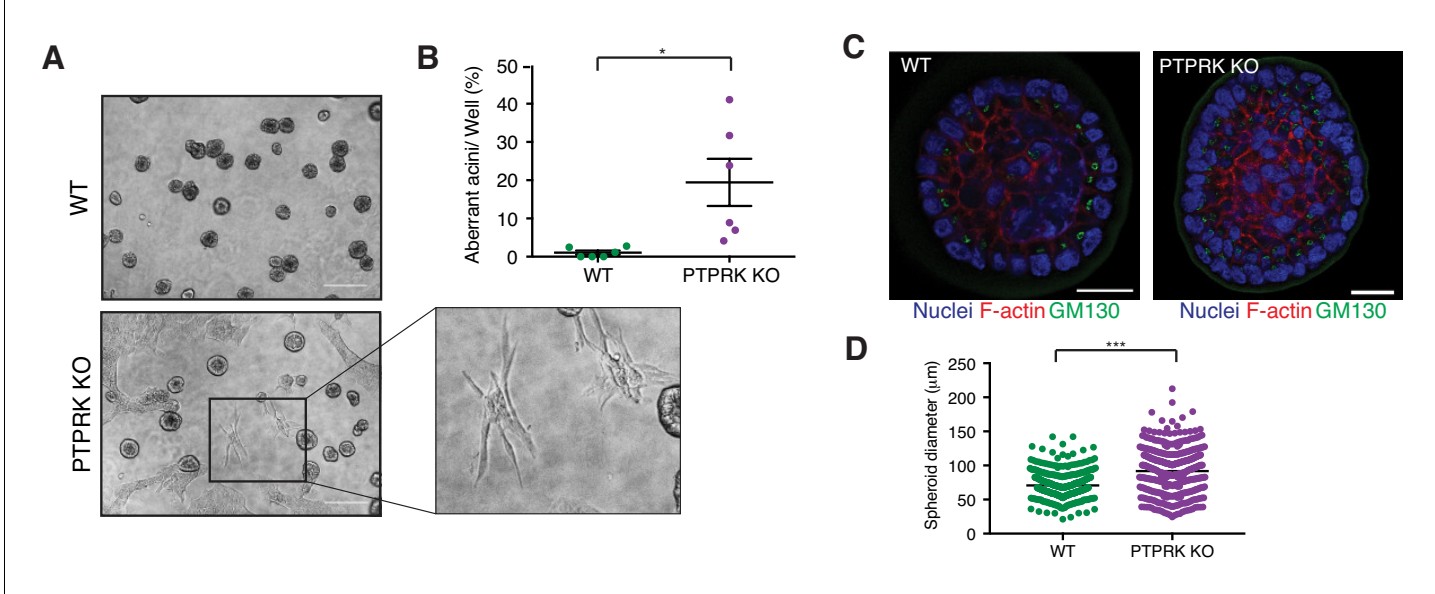

**Figure 8.** PTPRK promotes organization in epithelial cells. (**A**) Phase contrast images of wildtype and PTPRK KO. MCF10A spheroids after 14 day culture in Matrigel. Scale bar = 200 μm. (**B**) Frequency of aberrant acini observed in six independent wells each of wildtype and PTPRK KO MCF10A spheroids. Unpaired, two-tailed t test: *p<0.05. (**C**) Representative images of MCF10A spheroids stained for the Golgi marker GM130, F-actin and nuclei (Hoechst), after removal from Matrigel. Scale bar = 20 μm. (**D**) Circles were traced over cross sections, based on the Hoechst channel, for a total of 563 WT and 551 PTPRK KO immunostained spheroids from three entire slides per genotype and diameters calculated in Zen Pro. Unpaired, two-tailed t test: ***p<0.0005. See also *Figure 8—figure supplement 1*.

DOI: https://doi.org/10.7554/eLife.44597.027

The following source data and figure supplement are available for figure 8:

**Source data 1.** Source data used in graphs.
DOI: https://doi.org/10.7554/eLife.44597.029

**Figure supplement 1.** PTPRK loss perturbs epithelial organization.
DOI: https://doi.org/10.7554/eLife.44597.028

perhaps reflecting a limitation to the sensitivity or selectivity of our assays. Interestingly, these proteins were recently linked to mechanotransduction and hippo signaling, along with the PTPRK interactor MINK1 (*Meng et al., 2018*). An interesting future line of research will be to determine whether PTPRK is an upstream regulator of this new mechanotransduction pathway.

The cell junction organizers PARD3, Afadin and PLEKHA6 were all trapped by the C1089S mutant. Our immunoblots also highlighted phosphorylation-dependent enrichment of p120$^{Cat}$ and PKP4 on this trap. These five proteins were subsequently found to be hyperphosphorylated in PTPRK KO cells, suggesting that the C1089S mutant was effective in identifying substrates. However, we also found hyperphosphorylation of other PTPRK-binding partners in KO cells, indicating the traps alone were too restrictive in identifying substrates. Indeed, the PTPRK pseudophosphatase D2 domain was sufficient for substrate recognition, which may have masked the effect of the substrate traps, particularly as we have shown that D2 domain binding to most proteins is independent of tyrosine phosphorylation. By considering the entire PTPRK interactome, we could include the armadillo family proteins PKP3, PKP4 and p120$^{Cat}$ as substrates. Our criteria were quite conservative, and it is likely that ABLIM3, PLEKHA6, Girdin, KIAA1217 and PKP2 are also substrates, all of which are also linked to junction organization (*Gallegos et al., 2016*; *Guo et al., 2014*). Furthermore, we cannot rule out the existence of additional PTPRK substrates, for example, that are expressed in different cellular contexts, particularly as we observed divergent interactomes between epithelial and fibroblast cells.

Strikingly, most of the candidate PTPRK substrates identified here have orthologs in *Drosophila*, yet the R2B receptor family first appears in chordates (*Chen et al., 2017*). This suggests that rather than co-evolving with its substrates, PTPRK regulates pre-established functional protein complexes. In this way, PTPRK would have introduced new regulation, and perhaps function, to existing signaling networks for chordate and vertebrate-specific organization. Indeed, there are genetic links between orthologs for RAPGEF6, PARD3 and Afadin in the regulation of *Drosophila* AJ formation (*Bonello et al., 2018*). Furthermore, PARD3 and p120$^{Cat}$ *Drosophila* orthologs have been linked to the control of E-Cadherin internalization and recycling (*Bulgakova and Brown, 2016*). Several PTPRK substrates belong to the δ-catenin family, which undergoes significant expansion from one gene in *Drosophila* to seven in vertebrates (*Carnahan et al., 2010*). We did not detect ACRVF, PKP1 or δ-catenin (CTNND2) in MCF10A total proteomes (*Figure 3—source data 1*); however, these might be additional R2B family substrates in other cell types, such as neurons (*Paffenholz and Franke, 1997*). PKP2 was an interactor and hyperphosphorylated in PTPRK KO cells, but its dephosphorylation was not assessed. PKP3 has been proposed to promote the stability of desmosomes upon overexpression (*Gurjar et al., 2018*). PKP4 is targeted to both adherens junctions and desmosomes, but its role is less well understood (*Hatzfeld et al., 2003*). Our finding that PTPRK promotes junctional integrity raises the possibility that dephosphorylation of substrates, such as p120$^{Cat}$, would stabilize cadherin-based junctional assemblies. Interestingly, our ultrastructural analysis showed that PTPRK KO cells have leakier junctions and are shorter and less organized. This is reminiscent of the proposed role of p120$^{Cat}$ in controlling epithelial cell lateral domain expansion and shape maturation by balancing junctional contractility and maturation through regulation of E-Cadherin and RhoA (*Yu et al., 2016*), which reportedly depends on p120$^{Cat}$ tyrosine phosphorylation status (*Castaño et al., 2007*; *Davis et al., 2003*; *Fukumoto et al., 2008*). Indeed, we show that PTPRK dephosphorylates p120$^{Cat}$ in cells, and that PTPRK phosphatase activity is necessary to rescue junctional deficits.

Afadin appears to be a unique PTPRK substrate; it was not dephosphorylated by the very closely related PTPRM and it had the highest fold increase in tyrosine phosphorylation in PTPRK KO cells. Strikingly, this specificity is determined in large part by the PTPRK D2 pseudophosphatase domain, which was sufficient to recruit Afadin for dephosphorylation by the PTPRM D1 domain. This might reflect the greater identity between the PTPRK and PTPRM active D1 domains than the D2 domains (78% vs 73.6%). In line with this, we found little evidence for a PTPRK substrate consensus sequence other than a bias against basic residues immediately adjacent to the phosphotyrosine site, similar to previous reports for other PTPs (*Barr et al., 2009*). Several RPTPs have tandem intracellular PTP domains and the precise function of the inactive D2 domains remain to be determined (*Tonks, 2006*). The Janus kinases have a similar tandem arrangement where a pseudokinase domain regulates kinase activity (*Babon et al., 2014*). Indeed, regulation of the PTP D1 by D2 domains has been suggested for several RPTPs (*Toledano-Katchalski et al., 2003*). We do observe a three-fold reduction in PTPRK ICD enzyme activity compared to the D1 domain alone. However, when free D2 domain was added to free D1 domain, we saw no impact on activity. Instead, we find a key role for the pseudophosphatase domain in substrate recognition, similar to findings for CD45/PTPRC (*Nam et al., 2005*). We further show that unlike LAR (*Nam et al., 1999*), the PTPRK D2 domain could not easily be reactivated by mutation. Additionally, we rule out a role for the D2 domain in phosphotyrosine recognition using dephosphorylated lysates, vanadate competition and a PTPRK homology model. Several previous reports have shown that wildtype PTPs can interact with substrates, which are presumably in a dephosphorylated state (*Chen et al., 2006*; *Lee and Bennett, 2013*; *Timms et al., 1998*). Thus, PTPRK might serve as a scaffold for dephosphorylated proteins either by recruiting non-phosphorylated proteins, or by dephosphorylating already phosphorylated proteins upon recruitment. It is likely that the combination of recognition and dephosphorylation is important for full PTPRK function. Determining the spatiotemporal dynamics of PTPRK protein recruitment will be an important next step.

PTPRK and PTPRM both dephosphorylated PARD3, p120$^{Cat}$, PKP3 and PKP4 in lysates. Despite this, hyperphosphorylation of sites on each of these proteins were found in PTPRK KO cells, indicating that PTPRM cannot fully compensate for PTPRK loss. This could be due to lower PTPRM expression levels in MCF10A cells, possible differences in site selectivity or its intrinsically lower catalytic activity (this study and *Barr et al., 2009*). Vertebrate genomes all encode at least 4 R2B family members, with distinct expression profiles (*Figure 1—figure supplement 1A*). For example, by in situ

hybridization the receptors display divergent expression patterns in the adult mouse cerebellum (*Besco et al., 2004*). Assuming they are regulated similarly by cell-cell contact, our results indicate that receptor expression patterns will determine subtly distinct responses to cell contact.

Several phosphosites that are regulated by PTPRK have been characterized previously. p120$^{Cat}$-Y228 is phosphorylated in response to EGFR and a construct with N terminal phosphorylation-deficient mutations (including Y228F) is capable of rescuing adhesion phenotypes caused by p120$^{cat}$ deletion (*Mariner et al., 2004*). In contrast, a phosphomimetic p120$^{Cat}$-Y228E mutant increased recruitment of RhoA (*Castaño et al., 2007*). Afadin Y1237 phosphorylation, the rat equivalent of Human Afadin Y1230, has been shown to mediate recruitment of SHP2, implicating it in Ras-Mitogen activated protein kinase signaling (*Nakata et al., 2007*). This supports the role of PTPRK-mediated dephosphorylation of this site in tumor suppression.

PTPRK is the only R2B family member implicated by transposon-based mouse forward genetics in the progression of several cancers (*March et al., 2011*; *Starr et al., 2009*) and has been proposed to function as a tumor suppressor. Moreover, specific gene fusions result in its promoter driving the expression of oncogenic RSPO3 in a subset of colorectal cancers (*Seshagiri et al., 2012*). This is consistent with PTPRK being the predominant R2B receptor expressed in the mouse intestinal epithelium (*Haber et al., 2017*). Our findings of abrogated junction organization, spheroid overgrowth and invasive behavior in PTPRK-deficient cells support its role as a tumor suppressor. Several of the PTPRK substrates identified here have been linked to cancer, including PARD3 loss-of-function in invasion (*de la Rosa et al., 2017*), and oncogenic and tumor suppressive roles for p120$^{Cat}$ (*Schackmann et al., 2013*). Thus, their combined dysregulation could contribute to pathological phenotypes in a PTPRK mutant setting. Compromised epithelial barrier integrity is also linked to inflammatory bowel disease susceptibility; therefore, PTPRK SNPs linked to celiac disease should be investigated (*Trynka et al., 2011*). Our analysis of PTPRK KO cells showed downregulation of epithelial markers such as Keratin14, and upregulation of several mesenchymal markers such as vitronectin (VTN; *Figure 3C* and *Figure 3—source data 1*). PTPRK is a TGFβ target gene (*Wang et al., 2005*), and our data suggest it functions to suppress epithelial to mesenchymal transition (EMT), indicating a negative feedback role that could lead to pathological effects if perturbed (*Brabletz et al., 2018*).

Finally, our results provide evidence for cross-talk between PTPRK and growth factor signaling. Although PTPRK does not dephosphorylate EGFR in our assays, consistent with previous peptide assays (*Barr et al., 2009*), we did observe an interaction by BioID. Indeed, EGFR family interactions with R2B receptors have been reported (*Yao et al., 2017*). Growth factor stimulation leads to PTPRK tyrosine phosphorylation (*Batth et al., 2018*; *Reddy et al., 2016*) and it has been suggested that RTK-induced PTP inhibition by oxidation impacts cellular signaling (*Reynolds et al., 2003*). PTPs are well-placed to fine-tune and tailor responses to particular cellular contexts. Several PTPRK substrates are phosphorylated in a Src-dependent manner (*Reddy et al., 2016*). PTPRK might therefore provide feedback in the context of cell contact by dephosphorylating its substrates to promote junctional integrity. Indeed, overexpression of v-Src in epithelial cells leads to junction disassembly and EMT (*Woodcock et al., 2009*). Thus, such PTPs could act as interpreters of the cellular context. Interestingly, an analogous role for the contact-sensing RTK EphA2 was recently reported (*Stallaert et al., 2018*).

In summary, by defining the substrate repertoire of human PTPRK, we reveal mechanistic insight into its putative tumor suppressor role through its control of cell-cell junctions and suppression of EMT. Our study raises new questions about the phosphoregulation of junctional proteins and implicates PTPRK as a direct sensor and mediator of cell adhesion. We show that the PTPRK D2 domain is critical for substrate recognition, yet binds proteins independently of phosphorylation status. It will be of great interest to determine whether these findings hold true for other RPTPs. In addition, it is unknown whether the R2B receptor extracellular regions, which were previously described as spacer clamps (*Aricescu et al., 2007*), affect phosphatase activity or substrate recruitment. Finally, we show that PTPRK, like other PTPs (*Li et al., 2013*), does not recognize a peptide consensus sequence, unlike certain serine/threonine phosphatases (*Shi, 2009*), highlighting the need for the approach we have taken. Thus, we provide a framework for systematically identifying RPTP substrates, which in turn will advance our knowledge of these poorly characterized, yet important enzymes.

# Materials and methods

## Key resources table

| Reagent type (species) or resource | Designation | Source or reference | Identifiers | Additional information |
|---|---|---|---|---|
| Gene (*Homo sapiens*) | *PTPRK* | | ENSEMBL: ENST00000368213.9 | |
| Cell line (*H. sapiens*) | MCF10A | ATCC | CRL-10317 | |
| Cell line (*H. sapiens*) | HEK293T | D Ron | N/A | |
| Cell line (*H. sapiens*) | HEK293 | Sigma (ECACC) | 85120602-1VL | |
| Cell line (*H. sapiens*) | Hs27 Fibroblasts | Sigma (ECACC) | 94041901-1VL | |
| Cell line (*H. sapiens*) | MCF10A PTPRK KO A4 | This study | | CRISPR/Cas9 and clonal selection |
| Cell line (*H. sapiens*) | MCF10A PTPRK KO E3 | This study | | CRISPR/Cas9 and clonal selection |
| Cell line (*H. sapiens*) | MCF10A PTPRK KO H1 | This study | | CRISPR/Cas9 and clonal selection |
| Cell line (*H. sapiens*) | MCF10A PTPRK KO pooled | This study | | |
| Transfected construct (*H. sapiens*) | MCF10A PTPRK KO pooled.tGFP | This study | | Lentivirally transduced stable cell line |
| Transfected construct (*H. sapiens*) | MCF10A PTPRK KO pooled.tGFP.P2A.PTPRK | This study | | Lentivirally transduced stable cell line |
| Transfected construct (*H. sapiens*) | MCF10A PTPRK KO pooled.tGFP.P2A.PTPRK.C1089S | This study | | Lentivirally transduced stable cell line |
| Transfected construct (*H. sapiens*) | MCF10A.tGFP | This study | | Lentivirally transduced stable cell line |
| Transfected construct (*H. sapiens*) | MCF10A.tGFP.P2A.PTPRK.ECD-TMD.BirA*-Flag | This study | | Lentivirally transduced stable cell line |
| Transfected construct (*H. sapiens*) | MCF10A.tGFP.P2A.PTPRK.C1089S.BirA*-Flag | This study | | Lentivirally transduced stable cell line |
| Transfected construct (*H. sapiens*) | MCF10A PTPRK KO pooled.nuclear mApple | This study | | Lentivirally transduced stable cell line |
| Transfected construct (*H. sapiens*) | MCF10A.nuclear mApple | This study | | Lentivirally transduced stable cell line |
| Antibody | Rabbit monoclonal anti-PTPRK | This study | 2 .G6 | Western blot: 1:1000 |
| Antibody | Rabbit monoclonal anti-PTPRK | This study | 2 .H4 | Western blot: 1:1000 |
| Antibody | Rabbit monoclonal anti-PTPRK | This study | 2 .H5 | Western blot: 1:1000 |
| Antibody | Rabbit monoclonal anti-PTPRK | This study | 1 .F4 | FACS (1:200) and Immuno fluorescence (IF; 1:200) |
| Antibody | Mouse anti-PTPRK | Santa Cruz Biotechnology | Cat#Sc- 374315 | Western blot: 1:1000 (note: we did not observe any specific signal for PTPRK with this antibody) |
| Antibody | Rabbit anti-PARD3 | Sigma | Cat#HPA030443 (lot: C105765) | Western blot: 1:1000 |
| Antibody | Rabbit anti-PARD3 | Merck Millipore | Cat#07–330 | Western blot: 1:1000 |

*Continued on next page*

*Continued*

| Reagent type (species) or resource | Designation | Source or reference | Identifiers | Additional information |
|---|---|---|---|---|
| Antibody | Mouse anti-RAPGEF6 | Santa Cruz Biotechnology | Cat#sc-398642 (F-8) | Western blot: 1:1000 |
| Antibody | Mouse anti-Afadin | BD Transduction Labs | Cat#610732 | Western blot: 1:1000 |
| Antibody | Mouse anti-DLG5 | Santa Cruz Biotechnology | Cat#SC374594 (A-11) | Western blot: 1:1000 |
| Antibody | Mouse anti-PTPN14 | R and D Systems | Cat#MAB4458 | Western blot: 1:1000 |
| Antibody | Mouse anti-E-Cadherin | BD Transduction Labs | Cat#610181 | Western blot: 1:1000 IF: 1:100 |
| Antibody | Rabbit anti-b-Catenin | Cell Signaling Technology | Cat#9562S | Western blot: 1:1000 |
| Antibody | Rabbit anti-Phospho-EGFR (Y1068) | Cell Signaling Technology | Cat#3777S | Western blot: 1:1000 |
| Antibody | Rabbit anti-EGFR | Cell Signaling Technology | Cat#4267S | Western blot: 1:1000 |
| Antibody | Rabbit anti-phospho-tyrosine(P-Tyr-1000) | Cell Signaling Technology | Cat#8954 | Western blot: 1:2000 |
| Antibody | Rabbit anti-MAP4K4 | Cell Signaling Technology | Cat#5146 | Western blot: 1:1000 |
| Antibody | Rabbit anti-NUFIP2 | Bethyl Laboratories, Inc | Cat#A301-600A | Western blot: 1:1000 |
| Antibody | Rabbit anti-FMRP1 | ThermoFisher Scientific | Cat#MA5-15499 | Western blot: 1:1000 |
| Antibody | Rabbit anti-MINK1/MAP4K6 | ThermoFisher Scientific | Cat#PA5-28901 | Western blot: 1:1000 |
| Antibody | Rabbit anti-PKP4 | Bethyl Laboratories, Inc | Cat#A304-649A | Western blot: 1:1000 |
| Antibody | Mouse anti-P120 catenin | BD Transduction Laboratories | Cat#610133 | Western blot: 1:1000 IF: 1:100 |
| Antibody | Mouse anti-GM130 | BD Transduction Laboratories | Cat#610822 | Western blot: 1:1000 |
| Antibody | Rabbit anti-STAT3 | Cell Signaling Technology | Cat#4904S | Western blot: 1:1000 |
| Antibody | Rabbit anti-Paxillin | Cell Signaling Technology | Cat#12065 (D9G12) | Western blot: 1:1000 |
| Antibody | Mouse anti-Tubulin (Alpha) | Sigma | Cat#T6199 | Western blot: 1:1000 |
| Antibody | Mouse anti-PTPRM | Santa Cruz | Cat#sc-56959 | Western blot: 1:1000 |
| Antibody | Rabbit anti-PKP3 | Abcam | Cat#AB109441 | Western blot: 1:10000 |
| Antibody | Rabbit-anti-ABLIM3 | Sigma | Cat#HPA003245 | Western blot: 1:1000 |
| Antibody | Rabbit-Anti-ZO2 | ThermoFisher Scientific | Cat#711400 | Western blot: 1:1000 |
| Antibody | Rabbit-anti-Phospho-P120 catenin (Y904) | Cell Signaling Technology | Cat#2910 | Western blot: 1:1000 |
| Antibody | Rabbit-anti-Phospho-P120 catenin (Y228) | Cell Signaling Technology | Cat#2911 | Western blot: 1:1000 |
| Antibody | Rabbit polyclonal anti-Phospho-Paxillin (Y118) | Cell Signaling Technology | Cat#2541 | Western blot: 1:1000 |
| Antibody | Rabbit-anti-b-Actin | SIGMA | Cat#A2066 | Western blot: 1:1000 |
| Antibody | Mouse-anti-DSG3 | Bio-Rad | Cat#MCA2273T | Western blot: 1:5000 |
| Antibody | HRP conjugated-Donkey anti-Rabbit IgG | Jackson Immuno-Research | Cat#711-035-152 | Western blot: 1:5000 |

*Continued on next page*

*Continued*

| Reagent type (species) or resource | Designation | Source or reference | Identifiers | Additional information |
|---|---|---|---|---|
| Antibody | HRP conjugated- Donkey anti-Mouse IgG | Jackson Immuno-Research | Cat#711-035-152 | Western blot: 1:5000 |
| Antibody | HRP conjugated- Mouse anti-Rabbit IgG (Conformation specific) | Cell Signaling Technology | Cat#5127S | Western blot: 1:2000 |
| Antibody | Atto-488 Goat Anti-mouse IgG | Sigma | Cat#62197 | IF: 1:400 |
| Antibody | Atto-488 Goat Anti-mouse IgG | Sigma | Cat#62197 | IF: 1:400 |
| Antibody | Alexa Fluor-647 Goat Anti Rabbit IgG | Jackson Immuno-Research | Cat#111-605-003 | IF: 1:400 |
| Recombinant DNA reagent | pCW57.tGFP.P2A.MCS | Addgene | Cat#71783 | |
| Recombinant DNA reagent | pRK.HA.PTPRK.flag | Genentech | | Corresponds to Uniprot identifier: Q15262-3 |
| Recombinant DNA reagent | pRK.PTPRK(1-752).IgG1 | Genentech | | |
| Recombinant DNA reagent | pET15b | J. Deane | | |
| Recombinant DNA reagent | pSP.Cas9.(BB).eGFP | D Ron | | |
| Recombinant DNA reagent | pMD2.G | Addgene | Cat#12259 | |
| Recombinant DNA reagent | psPAX2 | Addgene | Cat#12260 | |
| Recombinant DNA reagent | pLenti-puro | Addgene | Cat#39481 | |
| Recombinant DNA reagent | PTPRK-BirA-R118G-Flag | A-C Gingras | | |
| Recombinant DNA reagent | pCW57.tGFP.P2A.PTPRK | This study | | |
| Recombinant DNA reagent | pCW57.tGFP.P2A.PTPRK.C1089S | This study | | |
| Recombinant DNA reagent | pCW57.tGFP.P2A.PTPRK(1-785).BirA-R118G.Flag | This study | | |
| Recombinant DNA reagent | pCW57.tGFP.P2A.PTPRK.C1089S.BirA-R118G.Flag | This study | | |
| Recombinant DNA reagent | pET15b.His.TEV.Avi | This study | | |
| Recombinant DNA reagent | pET15b.His.TEV.Avi.PTPRK.ICD | This study | | |
| Recombinant DNA reagent | pET15b.His.TEV.Avi.PTPRK.ICD.D1057A | This study | | |
| Recombinant DNA reagent | pET15b.His.TEV.Avi.PTPRK.ICD.C1089S | This study | | |
| Recombinant DNA reagent | pET15b.His.TEV.Avi.PTPRK.D1 | This study | | |
| Recombinant DNA reagent | pET15b.His.TEV.Avi.PTPRK.D2 | This study | | |

*Continued on next page*

*Continued*

| Reagent type (species) or resource | Designation | Source or reference | Identifiers | Additional information |
|---|---|---|---|---|
| Recombinant DNA reagent | pET15b.His.TEV.Avi. PTPRK.D2.triple | This study | | Mutations: A1346P, S1347D, L1384S, E1427Q, A1428T |
| Recombinant DNA reagent | pET15b.His.TEV .Avi.PTPRM.ICD | This study | | |
| Recombinant DNA reagent | pET15b.His.TEV .Avi.PTPRM.D1 | This study | | |
| Recombinant DNA reagent | pET15b.His.TEV. Avi. PTPRK-D1_K-D2. | This study | | |
| Recombinant DNA reagent | pET15b.His.TEV. Avi.PTPRM-D1_M-D2 | This study | | |
| Recombinant DNA reagent | pET15b.His.TEV. Avi.PTPRK-D1_M-D2. | This study | | |
| Recombinant DNA reagent | pET15b.His.TEV. Avi.PTPRM-D1_K-D2. | This study | | |
| Recombinant DNA reagent | pET15b.His.TEV .Avi.Src.sbSH2 | This study | | |
| Recombinant DNA reagent | pET15b.His.TEV. Avi.Grb2.sbSH2 | This study | | |
| Recombinant DNA reagent | pSP.Cas9.PTPRK.sgRNA1 | This study | | |
| Recombinant DNA reagent | pSP.Cas9.PTPRK.sgRNA2 | This study | | |
| Sequence-based reagent | ON-TARGETplus Human PTPRK siRNA | Dharmacon, GE Healthcare | Cat#J-004204–06 | |
| Sequence-based reagent | ON-TARGETplus Non-targeting pool siRNA | Dharmacon, GE Healthcare | Cat#D-001810-10-05 | |
| Sequence-based reagent | PTPRK CRISPR, BbsI.PTPRKgRNA1.Fwd | SIGMA | | CACCGCATGGATA CGACTGCGGCGG |
| Sequence-based reagent | PTPRK CRISPR, BbsI.PTPRKgRNA1.Rev | SIGMA | | AAACCCGCCGCA GTCGTATCCATGC |
| Sequence-based reagent | PTPRK CRISPR, BbsI.PTPRKgRNA2.Fwd | SIGMA | | CACCGATCTCGGG TGGTAGATAATG |
| Sequence-based reagent | PTPRK CRISPR, BbsI.PTPRKgRNA2.Rev | SIGMA | | AAACCATTATCTA CCACCCGAGATC |
| Sequence-based reagent | TaqMan probe: Hs02338565_gH (RPL19) | Thermo Fisher Scientific | Cat#4331182 | |
| Sequence-based reagent | TaqMan probe: Hs00267788_m1 (PTPRK) | Thermo Fisher Scientific | Cat#4331182 | |
| Sequence-based reagent | TaqMan probe: Hs00267809_m1 (PTPRM) | Thermo Fisher Scientific | Cat#4331182 | |
| Sequence-based reagent | TaqMan probe: Hs00179247_m1 (PTPRT) | Thermo Fisher Scientific | Cat#4331182 | |
| Sequence-based reagent | TaqMan probe: Hs00963911_m1 (PTPRU) | Thermo Fisher Scientific | Cat#4351372 | |
| Peptide, recombinant protein | DADE-pTyr-LIPQQG- phospho-peptide | Cambridge Research Biochemicals | Cat#crb1000746 | |
| Peptide, recombinant protein | END-pTyr-INASL- phospho-peptide | Cambridge Research Biochemicals | Cat#crb1000745 | |
| Peptide, recombinant protein | Catalase | Sigma | Cat#C134514 | |

*Continued on next page*

*Continued*

| Reagent type (species) or resource | Designation | Source or reference | Identifiers | Additional information |
|---|---|---|---|---|
| Peptide, recombinant protein | Cholera Toxin | Sigma | Cat#C-8052 | |
| Peptide, recombinant protein | Insulin | Sigma | Cat#I-1882 | |
| Peptide, recombinant protein | Epidermal Growth Factor | Peprotech | Cat#AF-100-15-1MG | |
| Peptide, recombinant protein | Lysyl endopeptidase (LysC) | Wako | Cat#129–02541 | |
| Peptide, recombinant protein | Trypsin (proteomics grade) | Thermo Fisher Scientific | Cat#90058 | |
| Commercial assay or kit | BIOMOL Green reagent | ENZO | Cat#BML-AK111-0250 | |
| Commercial assay or kit | Phosphate standard | ENZO | Cat#BML-KI102-0001 | |
| Commercial assay or kit | Q5 High-Fidelity DNA Polymerase | New England Biolabs | Cat#M0491S | |
| Commercial assay or kit | Phusion Hot Start II DNA polymerase | Thermo Fisher Scientific | Cat#F549L | |
| Commercial assay or kit | EZ-ECL substrate | Geneflow | Cat#K1-0170 | |
| Commercial assay or kit | NuPAGE MES (2-ethanesulfonic acid) SDS running buffer | ThermoFisher Scientific | Cat#NP0002 | |
| Commercial assay or kit | InstantBlue | Expedeon | Cat#ISB1L | |
| Commercial assay or kit | Phosphatase inhibitor cocktail | Roche | Cat#04906845001 | |
| Commercial assay or kit | TaqMan Universal Master Mix II | Applied Biosystems | Cat#4440040 | |
| Commercial assay or kit | MycoAlertTM PLUS Mycoplasma Detection Kit | Lonza | #LT07-705 | |
| Commercial assay or kit | MycoProbe Mycoplasma Detection Kit | R and D Systems | #CUL001B | |
| Chemical compound, drug | Hydrogen peroxide | Thermo Fisher Scientific | Cat#H/1750/15 | |
| Chemical compound, drug | Sodium orthovanadate | Alfa Aesar | Cat#J60191 | |
| Chemical compound, drug | 250 kDa-FITC-dextran | Sigma | Cat#FD250S-100MG | |
| Chemical compound, drug | Para-Nitrophenol-phosphate (pNPP) | New England Biolabs | Cat#P0757 | |
| Chemical compound, drug | IPTG | Generon | Cat#GEN-S-02122 | |
| Chemical compound, drug | D-biotin | Sigma | Cat#B4639 | |
| Chemical compound, drug | L-glutamine | Sigma | Cat#G7513 | |
| Chemical compound, drug | Hydrocortisone | Sigma | Cat#H-0888 | |
| Chemical compound, drug | Puromycin | Thermo Fisher Scientific | Cat#A11138-03 | |

*Continued on next page*

*Continued*

| Reagent type (species) or resource | Designation | Source or reference | Identifiers | Additional information |
|---|---|---|---|---|
| Chemical compound, drug | Phosphate free $H_2O$ | Thermo Fisher Scientific | Cat#10977–035 | |
| Chemical compound, drug | 8M Guanidine HCl | Thermo Fisher Scientific | Cat#24115 | |
| Chemical compound, drug | EPPS pH 8.5 | Alfa Aesar | Cat#561296 | |
| Chemical compound, drug | Trifluoroacetic Acid (TFA) | Thermo Fisher Scientific | Cat#28904 | |
| Chemical compound, drug | Acetonitrile | VWR | Cat#8364.290 | |
| Chemical compound, drug | Sodium phosphate dibasic ($Na_2HPO_4$) | Acros Organics | Cat#343811000 | |
| Chemical compound, drug | $NH_4OH$ | Acros Organics | Cat#460801000 | |
| Chemical compound, drug | Methanol-free 16% (w/v) parafor maldehyde (PFA) | Thermo Fisher Scientific | Cat#28906 | |
| Software, algorithm | Maxquant | Computational Systems Biochemistry | | Max Planck Institute of Biochemistry |
| Software, algorithm | Perseus | Computational Systems Biochemistry | | Max Planck Institute of Biochemistry |
| Software, algorithm | FIJI/ImageJ | Laboratory for Optical and Computational Instrumentation | | University of Wisconsin-Madison |
| Software, algorithm | Zen Blue | Zeiss | | |
| Software, algorithm | Zen Black | Zeiss | | |
| Software, algorithm | Graphpad | Prism | | |
| Software, algorithm | Chimera | UCSF | | |
| Other | HRP-conjugated Streptavidin | Thermo Fisher Scientific | Cat#434323 | |
| Other | STABLE competent *E. coli* | NEB | Cat#C3040I | |
| Other | DH5alpha competent *E. coli* | Invitrogen | Cat#18265017 | |
| Other | BL21 DE3 Rosetta *E. coli* | J Deane | N/A | |
| Other | DMEM | Thermo Fisher Scientific | Cat#41965–039 | |
| Other | Ham's F-12 | Sigma | Cat#N4888 | |
| Other | Horse Serum | Thermo Fisher Scientific | Cat#16050–122 | |
| Other | Fibroblast growth medium (FGM) | Promocell | Cat#C-23010 | |
| Other | Fetal Bovine Serum | Sigma | Cat#F7524-500ml | |
| Other | Trypsin-EDTA solution | Sigma | Cat#T3924 | |
| Other | GeneJuice transfection reagent | Merck Millipore | Cat#70967–3 | |
| Other | EDTA-free protease inhibitors | Roche | Cat#11836170001 | |
| Other | Lipofectamine RNAiMax | Invitrogen | Cat#13778075 | |

*Continued*

| Reagent type (species) or resource | Designation | Source or reference | Identifiers | Additional information |
|---|---|---|---|---|
| Other | OptiMEM | Thermo Fisher Scientific | Cat#31985070 | |
| Other | Lipofectamine LTX | ThermoFisher Scientific | Cat#15338100 | |
| Other | Protein G agarose beads | Merck Millipore | Cat#16–266 | |
| Other | Ni-NTA agarose | QIAGEN | Cat#1018244 | |
| Other | Streptavidin-coated magnetic beads | New England Biolabs | Cat#S1420S | |
| Other | Streptavidin agarose | ThermoFisher Scientific | Cat#20357 | |
| Other | DMEM SILAC media | Thermo Fisher Scientific | Cat#PI89985 | |
| Other | Ham's F-12 SILAC media | Thermo Fisher Scientific | Cat#88424 | |
| Other | Heavy Arginine + 10 | Sigma | Cat#608033–250 mg | |
| other | Heavy Lysine + 8 | Sigma | Cat#608041–100 mg | |
| Other | Proline | Sigma | Cat#P0380 | |
| Other | Light Arginine | Sigma | Cat#A5006 | |
| Other | Light Lysine | Sigma | Cat#L5501 | |
| Other | Hoechst 33342 | Thermo Fisher Scientific | Cat#62249 | |
| Other | BODIPY 558/568 phalloidin | Invitrogen | Cat#B3475 | IF: 1:400 |
| Other | ProLong Gold antifade | Invitrogen | Cat#P36934 | |
| Other | Normal Serum Block | BioLegend | Cat#927502 | |
| Other | Matrigel | Corning | Cat#356231 | |
| Other | 0.2 mm nitrocellulose membrane | GE Healthcare | Cat#15289804 | |
| Other | 0.4 mm pore size Transwell filter | Corning | Cat#353095 | |
| Other | 24-well companion plates for Transwell filters | Corning | Cat#353504 | |
| Other | Millicell ERS-2 Volt/Ohm meter | Merck Millipore | Cat#MERS00002 | |
| Other | Superdex 200 16/600 column | GE Healthcare | Cat#28-9893-35 | |
| Other | Superdex 75 16/600 column | GE Healthcare | Cat#28-9893-33 | |
| Other | Ultracel-3K regenerated cellulose centrifugal filter | Merck Millipore | Cat#UFC900324 | |
| Other | Ultracel-10 K regenerated cellulose centrifugal filter | Merck Millipore | Cat#UFC901024 | |
| Other | Ultracel-30 K regenerated cellulose centrifugal filter | Merck Millipore | Cat#UFC903024 | |
| Other | NuPAGE 4–12% Bis-Tris gel | Thermo Fisher Scientific | Cat#NP0321BOX | |
| Other | 1.5 ml low protein binding centrifuge tubes | Eppendorf | Cat#0030 108. 116 | |

*Continued on next page*

*Continued*

| Reagent type (species) or resource | Designation | Source or reference | Identifiers | Additional information |
|---|---|---|---|---|
| Other | 1cc/50 mg Sep-Pak Vac tC18 cartridges | Waters | Cat#WAT054960, | |
| Other | 1.5 ml Diagenode sonicator tubes | Diagenode | Cat#C30010010 | |
| Other | 5 ml low protein binding centrifuge tubes | Eppendorf | Cat#0030 108.302 | |
| Other | 2 ml low protein binding centrifuge tubes | Thermo Fisher Scientific | Cat#88379 | |
| Other | Graphite spin columns | Thermo Fisher Scientific | Cat#88302 | |
| Other | Titansphere Phos-TiO Tips (200 ml/3 mg) | GL Sciences Inc | Cat#5010–21311 | |
| Other | 18 mm x 18 mm,1.5 mm thick high-performance coverslips | Zeiss | Cat#474030-9000-000 | |

## Cells and cell culture

MCF10A cells were purchased directly from the American Type Culture Collection (ATCC; LGC Standards), and HEK293 and Hs27 cells were from the European Collection of Authenticated Cell Lines (ECACC; Sigma- Aldrich, UK). Cells were cultured in 75 cm$^2$ vented tissue culture flasks and incubated at 37°C in a humidified 5% $CO_2$ atmosphere and passaged, using trypsin-EDTA solution (Sigma-Aldrich), prior to reaching confluence, typically every 2–4 days depending on the cell line. MCF10A cells were grown in MCF10A growth media as described by the Brugge lab (*Debnath et al., 2003*) consisting of 50:50 DMEM (Thermo Fisher Scientific, UK)/Ham's F-12 (Sigma-Aldrich) containing 5% (v/v) horse serum (Thermo Fisher Scientific), 20 ng/ml EGF (Peprotech, UK), 0.5 µg/ml hydrocortisone (Sigma-Aldrich), 100 ng/ml cholera toxin (Sigma-Aldrich) 10 µg/ml insulin (Sigma-Aldrich). Hs27 cells were cultured in Fibroblast growth medium (Promocell, UK). HEK293 and HEK293T cells were cultured in DMEM containing 10% (v/v) FBS (Sigma-Aldrich), 2 mM L-glutamine (Sigma-Aldrich). Cell lines were tested for the presence of Mycoplasma using commercially available kits (see Key Resources table).

For SILAC analysis, MCF10A cells were cultured for 14 days in modified MCF10A growth media containing 50:50 SILAC DMEM (Thermo Fisher Scientific): SILAC F12 (Thermo Fisher Scientific), 5% (v/v) dialyzed horse serum and other supplements described above. For heavy labeling, 50 µg/ml lysine +8 (Sigma-Aldrich), 40 µg/ml arginine +10 (Sigma-Aldrich), 200 µg/ml proline (Sigma-Aldrich) were added. For light labeling, 50 µg/ml lysine (Sigma-Aldrich) and 40 µg/ml arginine (Sigma-Aldrich) were added. Isotopic labeling was assessed by mass spectrometry, following in-gel tryptic digest. At the start of each experiment heavy amino acid incorporation was ≥93%.

## Plasmids and constructs

Amino acid (aa) numbering is based on the following sequences; PTPRK; UniProt ID: Q15262-3, PTPRM; UniProt ID: P28827-1. All point mutations were introduced by polymerase chain reaction (PCR) using either Q5 High-Fidelity DNA (New England Biolabs, UK) or Phusion Hot Start II DNA (Thermo Fisher Scientific) polymerases as per manufacturer's protocol. The cDNA for the human PTPRK extracellular domain (ECD) of (aa 1–746) was synthesized with a C-terminal IgG1 tag fusion (GenScript, USA) and subcloned into the pRK vector (Genentech, USA). For transient mammalian expression, full-length human PTPRK coding expressing a N-terminal hemagglutinin (HA) tag and a C-terminal Flag tag was subcloned into the pRK vector. For stable integration with lentivirus infection, full length human PTPRK with and without a C-terminal BirA-R118G (BirA*)-Flag tag and truncated PTPRK (aa 1–785) with a C terminal BirA*-Flag tag were subcloned in-frame into pCW57. GFP.2A.MCS (a gift from Adam Karpf; #71783, Addgene, USA). For labeling nuclei, mApple with a C-terminal SV40 large T-antigen nuclear localization signal (PKKKRKV) was subcloned into the pLenti-puro vector (a gift from Ie-Ming Shih; #39481, Addgene). For bacterial expression, human

coding sequences corresponding to PTPRK D1 (aa 864–1150) PTPRK D2 (aa 1150–1439), PTPRK intracellular domain (ICD; aa 864–1439), PTPRK ICD-C1089S, PTPRK ICD-D1057A, PTPRM D1(aa 877–1163), PTPRM ICD (aa 877–1452), PTPRK D1 (aa 864–1147)-BstBI-PTPRK D2 (aa 1150–1439), PTPRM D1 (aa 877–1159)-BstBI-PTPRM D2 (aa 1160–1452), PTPRK D1 (aa 864–1147)-BstBI-PTPRM D2 (aa 1160–1452), PTPRM D1 (aa 877–1159)-BstBI-PTPRK D2 (aa 1150–1439) were subcloned into a modified pET-15b bacterial expression vector in frame encoding an N-terminal His.TEV.AviTag (MG SSHHHHHHSSGVDLGTENLYFQGTGGLNDIFEAQKIEWHEGGGS).

The previously described Src and Grb2 mutant SH2 domains (Bian et al., 2016) were synthesized (Thermo Fisher Scientific) and subcloned into the same modified pET-15b bacterial expression vector by restriction digest.

## Antibody production

New Zealand White (NZW) Rabbits were purchased from Western Oregon Rabbit Company (WORC). Rabbits were housed and immunized in Josman, LLC. The guideline of the animal care was under regulation of the Institutional Animal Care and User Committee (IACUC) requirement. The immunization protocol was approved by Roche IACUC and Genentech Laboratory Animal Resources. New Zealand White (NZW) rabbits were immunized with murine PTPRK protein. Rabbit anti-PTPRK mAb were generated from an antigen-specific single B cell cultivation and cloning platform based on a modified protocol (Seeber et al., 2014). PTPRK+/IgG + single B cells were directly sorted into culture plates using flow cytometry. The B cell culture supernatants were collected for High-Throughput screening by ELISA for binding to murine PTPRK and an unrelated control protein. PTPRK-specific B cells were lysed and immediately frozen at −80°C until molecular cloning. Variable regions (VH and VL) of each monoclonal antibody from rabbit B cells were then cloned into expression vectors from extracted mRNA as previously described (Seeber et al., 2014). Individual recombinant rabbit antibodies were expressed in Expi293 cells and subsequently purified with protein A. Purified anti-PTPRK antibodies were then subjected to functional activity assays and kinetic screening. Lead clones were selected for large scale antibody production.

## Lipid-based transfection of siRNA duplexes

Cells were transfected with siRNA duplexes using lipofectamine RNAiMAX (Thermo Fisher Scientific). For a 6-well plate, 15 µl of 2 µM siRNA duplexes were added to 481 µl of serum/antibiotic-free Opti-MEM (Thermo Fisher Scientific) and allowed to settle at RT for 5 min. 4 µl of lipofectamine RNAiMAX was then added, the mixture inverted briefly and incubated at RT for 20 min. Cells were seeded at $1.25–2.5 \times 10^5$ cells/ml in a 1 ml volume of complete growth medium, followed by immediate drop-wise addition of the siRNA/lipofectamine mixture to give a final siRNA concentration of 20 nM. Cells were returned to the incubator after 30 min at RT. After 24 hr total incubation, media were replaced for complete growth medium. Cells were allowed to recover for 48–72 hr prior to treatment or processing for analysis. All siRNA duplexes where purchased from Dharmacon (Horizon Discovery, UK).

## CRISPR/Cas9 genome editing

Oligos for single guide RNAs targeting exons 1 and 2 of PTPRK were cloned into pspCas9.(BB). eGFP as previously described (Ran et al., 2013). MCF10A cells were transfected with plasmids using Lipofectamine LTX with PLUS Reagent as per manufacturer's instructions (Thermo Fisher Scientific). After 48 hr eGFP positive cells were single-cell sorted using flow cytometry. Clones were expanded and protein levels assessed by Western blot. Targeted regions of the genome were amplified by PCR and sequenced to confirm editing.

## Lentivirus production and infection

$15 \times 10^6$ HEK293T cells were seeded in 12 ml of complete growth medium/15 cm$^2$ dish (two dishes per lentivirus) and incubated for 24 hr at 37°C with 5% CO$_2$. Each 15 cm$^2$ dish was then transfected with either 6 µg of pCW57.GFP.2A. or pLenti.puro expression plasmid encoding the desired construct, 12 µg of the psPAX2 packing plasmid (a gift from Didier Trono; #12260, Addgene) and 3 µg of the pMD2.G envelope plasmid (a gift from Didier Trono; #12259, Addgene) using the GeneJuice transfection reagent (Merck Millipore, UK) as per manufacturer's instructions. After 24 hr media was then replaced with 16 ml complete growth medium. 48–72 hr post-transfection, culture medium was

collected and filtered through a 0.45 µm mixed cellulose esters membrane. Viral particles were pelleted via ultracentrifugation at 100,000 x g for 1.5 hr at 4°C and resuspended in 600 µl of OptiMEM (Thermo Fisher Scientific). Lentivirus was aliquoted and stored at −80°C until required.

For lentiviral infections, $1.6 \times 10^5$ cells were seeded per well of a six well plate in 900 µl of growth medium, prior to the drop-wise addition of 100 µl lentivirus. After 30 min at room temperature (RT), cells were returned to the incubator. 72 hr later cells were reseeded in 0.4 µg/ml puromycin (Gibco, Thermo Fisher Scientific) selection medium.

### PTPRK extracellular domain screen

PTPRK ECD was expressed in HEK293S cells and purified using standard affinity chromatography procedures. Purified recombinant PTPRK ECD was screened as protein A microbeads complexes, carrying a Cy5-labeled IgG as an inert carrier to allow visualization of any binding partners against the Extracellular Protein Microarray Technology, as described previously (*Martinez-Martin et al., 2016*; *Yeh et al., 2016*). This platform (consisting of >1500 purified proteins, representing ≈50% of the single transmembrane-containing receptors in humans), in combination with a query protein multimerization approach for enhanced detection of binding partners, has enabled identification of multiple interactions between extracellular proteins (*Martinez-Martin et al., 2016*; *Yeh et al., 2016*), including low affinity interactions that often characterize receptors expressed on the cell surface (*Martinez-Martin, 2017*; *Wright, 2009*).

### Pervanadate treatment

Two $\times 10^6$ cells were seeded per $10 \text{ cm}^2$ dish and cultured for 6 days with a media change on day 3 and day 5. Cells were stimulated with 6 ml of complete growth medium containing 1 mM fresh sodium pervanadate (made as outlined below) for 30 min at 37°C/5% $CO_2$. Cells were then transferred onto ice and washed twice with ice-cold PBS, prior to the addition of 600 µl of ice-cold lysis buffer (50 mM Tris-HCl pH 7.5, 150 mM NaCl, 10% (v/v) glycerol, 1% (v/v) triton X-100, 1 mM EDTA, 5 mM iodoacetamide, 1 mM sodium orthovanadate, 10 mM NaF, 1X EDTA-free protease inhibitors (Roche, UK)), and incubated on a rocker at 4°C in the dark. Lysates were harvested, followed by the addition of DTT to a final concentration of 10 mM and incubated for 15 min on ice. Lysates were cleared by centrifugation at 14000 x g for 15 min at 4°C and supernatants were transferred into fresh tubes. Pervanadate-treated cell lysates were then snap frozen and stored at −80°C until required.

To generate a 50 mM pervanadate working stock, 5 µl of 3% (w/v) $H_2O_2$ (Thermo Fisher Scientific) was diluted in 45 µl of 20 mM HEPES pH 7.3 prior to the addition of 490 µl of 100 mM $Na_3VO_4$ (Alfa Aesar, Thermo Fisher Scientific) and 440 µl of $H_2O$, the solution was mixed by gentle inversion and incubated at RT for 5 min. After 5 min, a small amount of catalase (Sigma-Aldrich) was added to the pervanadate solution using a pipette tip and mixed by gentle inversion to quench unreacted $H_2O_2$. Freshly made pervanadate solution was used within 5 min to avoid decomposition of the complex.

### RT-qPCR

RNA was extracted using the RNeasy Plus Mini Kit (Qiagen, UK) according to the manufacturer's instructions. cDNA was prepared using the High-Capacity cDNA Reverse Transcription Kit as per manufacturer's instructions (Applied Biosystems). RT-qPCR was performed using the TaqMan Universal Master Mix II (Applied Biosystems, Thermo Fisher Scientific), 50 ng cDNA and specific Taqman probes for PTPRK, PTPRM, PTPRT, PTPRU and RPL19 Real-time PCR was performed with the 7900HT Fast Real-Time PCR System (Thermo Fisher Scientific). Expression levels were normalized to the reference gene RPL19. Gene specific primers are listed in the Key Resources table.

### SDS-PAGE and immunoblotting

SDS PAGE and immunoblotting were carried out as previously described (*Fearnley et al., 2015*). 25–50 µg of cell lysate was resuspended in an appropriate volume of 5X SDS-PAGE sample buffer (0.25 M Tris-HCl pH 6.8, 10% (w/v) SDS, 20% (v/v) glycerol, 0.1% (w/v) bromophenol blue, 10% (v/v) β-mercaptoethanol) and incubated at 92°C for 5 min. Samples were run on a 8, 10 or 12% (v/v) SDS-polyacrylamide resolving gel with a 5% (v/v) SDS-PAGE stacking gel and subjected to electrophoresis at 120–130 V for ~1–2 hr in 25 mM Tris, 190 mM glycine, 0.1% (w/v) SDS. Proteins were transferred onto 0.2 µm reinforced nitrocellulose membranes (GE Healthcare) at 300 mA for 3–4 hr at 4°C

in 25 mM Tris, 190 mM glycine, 20% (v/v) methanol. Membranes were briefly rinsed in TBS-T (20 mM Tris pH 7.6, 137 mM NaCl, 0.1% (v/v) Tween-20) prior to incubation for 20–60 min in 5% (w/v) skimmed milk/TBS-T to block non-specific antibody binding. The blocking solution was removed and membranes rinsed in TBS-T prior to primary antibody incubation (4–5 hr at RT or overnight at 4°C). Membranes were then subjected to 3 × 10 min washes in TBS-T, prior to incubation with HRP-conjugated species-specific anti-IgG antibodies (1–2 hr at RT). Membranes were then subjected to 3 × 10 min washes in TBS-T, prior to being incubated with combined EZ-ECL solution (Geneflow, UK) and imaged using a Bio-Rad ChemiDoc MP imaging system.

## Expression, biotinylation (AviTag) and purification of recombinant proteins

*Escherichia coli* BL21(DE3) Rosetta cells transformed with the relevant expression construct were cultured at 30°C/220 rpm in 1 l of 2XTY medium containing 50 µg/ml carbenicillin and 34 µg/ml chloramphenicol until the OD600 reached 0.6–0.7. Cultures were then transferred to 20°C/220 rpm and allowed to equilibrate, prior to the addition of 1 mM isopropyl-thio-β-D-galactopyranoside (IPTG; Generon, UK) and 200 µM of D-biotin (Sigma-Aldrich). Cells were harvested after 20 hr by centrifugation at 4000 x g for 30 min and bacterial pellets stored at −20°C until required. Prior to lysis, cells were subjected to one round of freeze-thaw. Cells were lysed in purification buffer (50 mM HEPES pH 7.5 for PTP domains (50 mM Tris pH 7.4 for SH2 domain mutants), 500 mM NaCl, 5% (v/v) glycerol and 0.5 mM TCEP), containing EDTA-free protease inhibitor tablets (Roche) using a Constant Systems cell disruptor and the cell extract was clarified via centrifugation at 40000 x g for 30 min at 4°C. The supernatant was removed and incubated with 0.5 ml of Ni-NTA agarose (Qiagen) for 1 hr at 4°C. Ni-NTA Agarose was then pelleted via centrifugation at 500 x g for 5 min at 4°C and packed into a gravity flow column. Ni-NTA agarose was then washed with 10 volumes of purification buffer containing 5 mM imidazole, followed by 20 volumes of purification buffer containing 20 mM imidazole; prior to elution in purification buffer containing 250 mM imidazole. The eluted protein was then subjected to size exclusion chromatography (SEC) using a Superdex 200 16/600 column (GE Healthcare Life Sciences, Thermo Fisher Scientific) for PTP domains or Superdex 75 16/600 column (GE Healthcare Life Sciences, Thermo Fisher Scientific) for SH2 mutant domains. Columns were equilibrated in SEC buffer (50 mM HEPES pH 7.5 (50 mM Tris pH 7.4 for SH2 domains), 150 mM NaCl, 5% (v/v) glycerol, 5 mM DTT). Protein was concentrated to 2–10 mg/ml using an Ultracel-3K, Ultracel-10 K or Ultracel-30 K regenerated cellulose centrifugal filter (Merck Millipore), prior to snap-freezing and storage at −80°C until required. The purified protein was assessed by SDS-PAGE and staining with InstantBlue (Expedeon, UK).

## Confirmation of AviTag biotinylation via streptavidin gel shift assay

Biotinylated recombinant proteins (2–10 µg) were solubilized in 4 µl of 5X SDS-PAGE sample buffer and incubated at 95°C for 5 min. Samples were then cooled to RT and allowed to equilibrate for 5 min. 24 µl of 2 mg/ml streptavidin/PBS (approx. 5-fold molar excess) was then added and the mixture was incubated at RT for 5 min. Samples were then run on a NuPAGE 4–12% Bis-Tris gel (Thermo Fisher Scientific) in NuPAGE MES (2-ethanesulfonic acid) gel running buffer (Thermo Fisher Scientific) at 190 V for 30 min. Protein only and streptavidin only controls should be included. Proteins were then visualized via staining with InstantBlue (Expedeon) for 1 hr at RT. Gels were imaged using a Bio-Rad ChemiDoc MP imaging system and the percentage of biotinylated protein determined via 2D-densitometry using Fiji (*Schindelin et al., 2012*).

## Recombinant protein pull downs

25–50 µg (tandem or single domain, respectively) of biotinylated His.TEV.Avi.PTPx domains were conjugated to 167 µl of pre-washed streptavidin-coated magnetic beads suspension (4 mg/ml; New England Biolabs) in 500 µl of ice-cold size exclusion buffer (50 mM HEPES pH 7.5 (50 mM Tris pH 7.4 for SH2 domains), 150 mM NaCl, 5% (v/v) glycerol, 5 mM DTT) at 4°C for 1–2 hr on a rotator. A beads-only control was treated identically. Samples were briefly spun, transferred onto a magnetic stand and washed 3 times with 1 ml of ice-cold size exclusion buffer, followed by two washes with 1 ml of ice-cold 150 mM NaCl wash buffer (20 mM Tris-HCl pH 7.4, 150 mM NaCl, 10% (v/v) glycerol, 1% (v/v) triton X-100, 1 mM EDTA pH 8.0). Conjugated PTP domains were then blocked in 1 ml of

ice-cold 5% (w/v) BSA in 150 mM NaCl wash buffer containing 1x EDTA-free protease inhibitors (Roche) at 4°C for 1 hr on a rotator. Simultaneously, freshly thawed pervanadate-treated cell lysate was then pre-cleared with streptavidin-coated magnetic beads (167 µl of bead suspension (4 mg/ml) per ml of lysate) at 4°C for 1 hr on a rotator. Blocked conjugated PTPx domains were then briefly spun, transferred onto a magnetic stand and washed twice with 1 ml of ice-cold 150 mM NaCl wash buffer; prior to incubation with 1 ml of 1 mg/ml pre-cleared pervandate-treated lysate at 4°C on for 1.5 hr on a rotator. In a cold room, beads were pulled to a magnet and supernatant removed. Beads were then washed twice in 1 ml ice-cold 150 mM NaCl wash buffer including a brief spin and separation by magnet. Beads were then washed once with 1 ml ice-cold 150 mM NaCl wash buffer without resuspension and washed twice more in 1 ml ice-cold 150 mM NaCl wash buffer with resuspension. Next beads were washed once without resuspension and twice with resuspension in 1 ml ice-cold 500 mM NaCl wash buffer (20 mM Tris-HCl pH 7.4, 150 mM NaCl, 10% (v/v) glycerol, 1% (v/v) triton X-100, 1 mM EDTA pH 8.0). Finally, beads were washed once without resuspension and once with resuspension in 1 ml ice-cold TBS (20 mM Tris pH 7.6, 137 mM NaCl). For immunoblot analysis, beads were resuspended in 20 µl of 18% (v/v) formamide,1 mM EDTA pH 8.0 made up in TBS, incubated at 95°C for 5 min, followed by addition of 30 µl of 5x SDS-PAGE sample buffer containing 2 mM biotin and incubated at 95°C for 10 min. After a brief spin, beads were separated by magnet and supernatants subjected to SDS-PAGE. For analysis by mass spectrometry, beads were subject to two further washes without resuspension and one further wash with resuspension in 1 ml ice-cold 50 mM ammonium bicarbonate pH 8.0, followed by on-bead tryptic digest.

## On-bead tryptic digest

Streptavidin beads for tryptic digest were resuspended in 95 µl of 50 mM ammonium bicarbonate pH 8.0, prior to the addition of 5 µl of 100 mM DTT (5 mM final DTT concentration), and incubation at 56°C for 30 min. 10 µl of 154 mM iodoacetamide (IAA) was then added (14 mM final IAA concentration) and samples incubated in the dark at RT for 20 min. Unreacted IAA was then quenched by the addition of 7 µl of 100 mM DTT (10 mM final DTT concentration), and incubation at RT for 15 min. Next, 31.5 µl of 50 mM ammonium bicarbonate pH 8.0 and 1.5 µl of LysC (0.005 AU/µl; Wako) was added to each sample, followed by incubation at RT for 3 hr with shaking. 150 µl of 7.7 ng/µl trypsin (Thermo Fisher Scientific) in 50 mM ammonium bicarbonate pH 8.0) was added to each sample (3.84 ng/µl final trypsin concentration) and incubate at 37°C overnight with shaking. An additional 150 µl of 7.7 ng/µl trypsin was then added to each sample (5.1 ng/µl final trypsin concentration), followed by incubation at 37°C for 2 hr with shaking. Samples were briefly spun and placed onto a magnetic stand, supernatant was then transferred into a low protein-binding tube (Eppendorf, Thermo Fisher Scientific). Beads were then washed twice with 150 µl of proteomics grade water (Thermo Fisher Scientific) and resulting supernatants added to the first supernatant. Samples were then centrifuged at 18400 x g for 10 min at 4°C and supernatant transferred into a new low protein-binding tube. Samples were then adjusted to 1% (v/v) TFA, prior to centrifugation at 21000 x g for 10 min at 4°C and supernatants transferred into a new low protein-binding tube. Each tryptic digest was desalted using a 1cc/50 mg Sep-Pak C18 cartridge (Waters). All buffers were made using proteomics grade water. Sep-Paks were equilibrated via washing with twice with 1 ml 100% (v/v) acetonitrile (AcN; VWR), twice with 1 ml 50% (v/v) AcN/0.1% (v/v) TFA and twice with 1 ml 0.1% (v/v) TFA. Samples were then slowly loaded onto each Sep-Pak; flow-through was reapplied once. Sep-Paks were then washed three times with 1 ml 0.1% (v/v) TFA. Peptides were then eluted into a new low protein-binding tube by addition of two 350 µl volumes of 50% (v/v) AcN/0.1% (v/v) TFA. Peptide samples were then dried down using a vacuum centrifuge (Concentrator 5301, Eppendorf) at 30–45°C. Peptide pellets were then stored at −20°C until further processing.

## Mass spectrometry acquisition and data analysis for pull downs

LC-MS/MS data were acquired on either a Q Exactive (Thermo Fisher Scientific) or a Q Exactive Plus (Thermo Fisher Scientific) each coupled, via an EASYspray source, to an RSLC3000 nanoUHPLC. Peptides were loaded onto a 100 µm ID x 2 cm Acclaim PepMap nanoViper precolumn (Thermo Fisher Scientific) and resolved using a 75 µm ID x 50 cm, 2 µm particle PepMap RSLC C18 EASYspray column at 40°C. NanoUHPLCs were operated with solvent A (0.1% formic acid) and solvent B (80% MeCN, 0.1% formic acid). Peptides were resolved on the Q Exactive by a gradient rising from 3% to

40% B by 60 mins and on the Q Exactive Plus by a gradient ring from 10% to 40% B by 57 min. MS spectra on the Q Exactive were acquired between m/z 400 to 1400 and between m/z 400 to 1500 on the Q Exactive Plus. Both operated MS/MS triggered in a top 10 DDA fashion.

Raw files were processed on MaxQuant v.1.5.2.8 or 1.5.8.3. using default settings. Quantification was carried out using Perseus ver. 1.5.8.5 (*Tyanova et al., 2016*). For label-free quantification (LFQ), LFQ intensities from MaxQuant were log2(x) transformed prior to filtering out proteins branded as identified only by site, reverse or potential contaminants. Proteins were then further filtered out based on the minimum number of valid values in one group, to be stringent we required a minimum of three (MCF10A experiments) or two (Hs27 experiments) valid values. Missing values were then imputed from the normal distribution and statistical significance was calculated via a two-sample, two-sided t test performed with truncation by a permutation-based FDR (threshold value 0.05). High confidence interactors were defined as >2 fold enrichment (over beads only), significant (p>0.05) and a CRAPome score ≤137 (*Mellacheruvu et al., 2013*).

## pNPP phosphatase activity assay

All buffers were made in phosphate-free $H_2O$ (Thermo Fisher Scientific). Recombinant phosphatase was added to a 96-well plate in a total volume of 50 µl reaction buffer (50 mM HEPES pH 7.4, 150 mM NaCl, 5% (v/v) glycerol, 5 mM DTT). 50 µl of reaction buffer containing 20 mM pNPP (New England Biolabs) was then added and the plate was incubated at RT for 3–15 min. Reactions were stopped by the addition of 50 µl 0.58 M NaOH (0.193 final concentration) and the absorbance read at 405 nm using a 96-well plate reader (SpectraMax M5, Molecular Devices, UK).

## BIOMOL green phosphatase activity assay

All buffers were made in phosphate-free water (Thermo Fisher Scientific). In a 96-well plate, 30 µl of reaction buffer containing 100 µM each of DADE-pTyr-LIPQQG-Acid phosphopeptide and END-pTyr-INASL-Acid phosphopeptides (Cambridge Research Biochemicals) was incubated at 30°C for 3 min, prior to the addition of recombinant phosphatase in a total volume of 20 µl reaction buffer. The assay was then incubated at RT for 2.5–3.5 min. Reaction was stopped by the addition of 100 µl of BIOMOL Green reagent (ENZO, UK), followed by incubation at RT for 15–30 min. The absorbance was then read at 620 nm using a 96-well plate reader (SpectraMax M5, Molecular Devices). Enzyme activity was compared against a standard curve from serial dilutions of a phosphate standard (ENZO).

## Quantitative tyrosine phosphoproteomics and total proteomics

$2 \times 10^6$ WT or PTPRK-KO SILAC labeled MCF10A cells were seeded into three 10 $cm^2$ dishes in heavy (WT) or light (PTPRK-KO) SILAC medium for each experiment. Cells were cultured for 7 days with a media change on days 2 (10 ml), 4 (12 ml), 5 (12 ml) and 6 (12 ml). On day 7, cells were placed on ice, washed twice with ice-cold PBS and lysed in 150 µl of 6M guanidine (Thermo Fisher Scientific) in 50 mM EPPS pH 8.5 (Alfa Aesar) with 1X EDTA-free protease inhibitor cocktail (Roche) and 1X phosphatase inhibitor cocktail (Roche). Samples were transferred into Diagenode sonication tubes (Diagenode, UK) on ice, vortexed at max speed for 30 s and sonicated at 4°C on high power for 5 × 30 s pulses using a water bath sonicator (Bioruptor, Diagenode). Samples were cleared twice by centrifugation at 13000 x g for 10 min at 4°C with supernatants transferred to new low protein-binding tubes (Eppendorf). Protein concentration was then determined by BCA assay and equal amounts of heavy and light labeled protein lysates were transferred into 5 ml low protein-binding tubes (Eppendorf) to give a maximum combined volume of 600 µl. A total of 10 mg of protein from heavy and light lysates was processed per replicate. Proteins were reduced by addition of 30 µl DTT/200 mM EPPS pH 8.5 (5 mM final DTT concentration), vortexed and incubated at RT for 20 min. Proteins were then alkylated by addition of 16.8 µl of 500 mM IAA/200 mM EPPS pH 8.5 (14 mM final IAA concentration), vortexed and incubated at RT for 20 min in the dark. Unreacted IAA was quenched via the addition of 30 µl of freshly thawed 100 mM DTT/200 mM EPPS pH 8.5 (8.9 mM final DTT concentration), prior to vortexing and incubation at RT for 15 min. Samples were diluted to a final concentration of 1.5 M guanidine by addition of 1.8 ml 200 mM EPPS pH 8.5. Next, 0.06 AU of LysC (Wako) was added to each sample, prior to vortexing and incubation at RT for 3 hr with shaking. Samples were split in half and transferred to two new 5 ml low protein-binding tubes. Samples were

diluted to a final concentration of 0.5 M Guanidine by adding 2.48 ml 200 mM EPPS pH 8.5. 100 µl of 124 ng/µl Trypsin (Thermo Fisher Scientific)/EPPS pH 8.5 was then added to each sample, prior to vortexing and incubation at 37°C overnight with shaking. An additional 100 µl of 124 ng/µl Trypsin/ EPPS pH 8.5 was then added, prior to vortexing and incubation at 37°C for 2 hr with shaking. Tryptic digests were then acidified via the addition of 39.8 µl TFA (Thermo Fisher Scientific) or 1% (v/v) TFA final concentration. Samples were then split into two new 2 ml low protein-binding tubes (Thermo Fisher Scientific), prior to centrifugation at 21000 x g for 10 min. Supernatants were transferred to a new 2 ml low protein-binding tubes, prior to being snap-frozen and stored at −80°C or desalted. Tryptic digests were desalted using 1cc/50 mg Sep-Pak Vac tC18 cartridges (Waters, UK); 20 mg/ ~40 ml of tryptic digest was split across four 1cc/50 mg Sep-Pak Vac tC18 cartridges. Sep-Paks were equilibrate, washed and loaded as described above. Peptides were eluted in a stepwise manner into new 1.5 ml low protein -binding tubes. Fraction 1: 350 µl 12.5% (v/v) AcN/0.1% (v/v) TFA, Fraction 2: 350 µl 25% (v/v) AcN/0.1% (v/v) TFA, Fraction 3: 350 µl 37.5% (v/v) AcN/0.1% (v/v) TFA, Fraction 4: 350 µl 50% (v/v) AcN/0.1% (v/v) TFA. Corresponding fractions were then pooled and 10% (v/v) removed for total proteome analysis. Peptides were then dried down using a vacuum centrifuge (Concentrator 5301, Eppendorf) at 45°C and stored at −20°C until further processing.

For phospho-tyrosine enrichment, peptide fractions were resuspended in 400 µl of ice-cold IAP buffer (50 mM Tris-HCL pH 7.4, 10 mM Na$_2$HPO$_4$ (Acros Organics), 100 mM NaCl) and incubated for 10 min on ice. Added to each fraction was 10 µl of rabbit anti-pY-1000 antibody (Cell Signal Technologies, New England Biolabs) pre-conjugated to 5 µl of protein G agarose bead suspension (Merck Millipore) and 2.4 µg each of biotinylated Src and Grb2 SH2 mutant domains, pre-conjugated to 5 µl of streptavidin agarose bead suspension (Thermo Fisher Scientific) and ice-cold IAP buffer up to 1 ml. Samples were then incubated at 4°C for 16–24 hr on a rotator. Beads were pelleted via at 14000 x g for 30 s and washed three times with 1 ml ice-cold IAP buffer followed by two washes with 1 ml ice-cold proteomics grade water. Peptides from each fraction were eluted in 125 µl of 0.15% (v/v) TFA at RT for 15 min; beads were pelleted and the supernatant transferred to a new 1.5 ml low protein binding tube (Eppendorf). This step was repeated for a total of three elutions and supernatants combined. Eluted peptides were then desalted using graphite spin columns (Thermo Fisher Scientific), according to manufacturer's instructions, using two columns per fraction, and dried down using a vacuum centrifuge at 45°C. For further enrichment of phospho-peptides using TiO$_2$, peptide fractions were resuspended in 100 µl 2% (v/v) TFA and incubated at RT for 10 min. Each fraction was then split and processed on two Titansphere Phos-TiO Tips (200 µl/3 mg; GL Sciences Inc) as per manufacturer's instructions. Peptides were eluted in 50 µl of 5% (w/v) NH$_4$OH (35% w/v; Acros Organics, Thermo Fisher Scientific), followed by 50 µl of 60% (v/v) AcN. Peptide samples were then dried down using a vacuum centrifuge at 45°C, prior to storage at −20°C or −80°C before analysis by mass spectrometry.

## Mass spectrometry acquisition and data analysis for quantitative tyrosine phosphoproteomics and total proteomics

Samples were resuspended in 20 µL sample solution (3% MeCN, 0.1% trifluoracetic acid). LC-MS/MS data acquisition was performed on a Q Exactive Plus and an Orbitrap Fusion Lumos (Thermo Fisher Scientific) with both instruments configured to RSLC3000 nanoUHPLCs. Both the Q Exactive Plus and the Fusion Lumos were operated with an EASYspray source using a 50 cm PepMap EASYspray emitter at 40°C. The Fusion Lumos was also operated using a 75 cm Acclaim PepMap column at 55°C with SilicaTip coated emitters (New Objective, USA). All nanoHPLCs were operated with solvent A (0.1% formic acid) and solvent B (80% MeCN, 0.1% formic acid).

Total peptides were resolved using four different gradients. Gradient 1 (for sample fraction 1) rose from 3% to 15% solvent B by 125 min and 40% B by 175 min. Gradient 2 (for sample fraction 2) rose from 3% to 25% B by 125 min and 40% B by 175 min. Gradient 3 (for sample fraction 3) rose from 3% to 40% B by 175 min and gradient 4 (for sample fraction 4) rose from 12% to 58% B by 175 min.

Phosphopeptides were resolved using four different gradients. Gradient 1 (for sample fraction 1) rose from 3% to 15% solvent B by 70 min and 40% B by 95 min. Gradient 2 (for sample fraction 2) rose from 3% to 25% B by 80 min and 40% B by 95 min. Gradient 3 (for sample fraction 3) rose from 10% to 40% B by 95 min and gradient 4 (for sample fraction 4) rose from 15% to 50% B by 95 min.

MS/MS data on the Q Exactive Plus were acquired in a Top10 DDA fashion and on the Fusion Lumos MS/MS data were acquired in the ion trap using a 3 s cycle.

Data were processed using MaxQuant v.1.6.2.3 with a Uniprot Homo sapiens database (downloaded 28/1/2018). Variable modifications were set as oxidation (M), acetylation (protein N-terminus) and phospho (STY) with 're-quantify' and 'match between runs' enabled. Peptide and protein FDR were set to 0.01. Quantification was carried out using Perseus ver. 1.5.8.5 (*Tyanova et al., 2016*). Normalized H/L ratios from MaxQuant were log2(x) transformed prior to filtering out proteins labeled as identified only by site or reverse. Proteins were then further filtered out based on the minimum number of valid values; a minimum of two valid values were required for high confidence analysis. Missing values were then imputed from the normal distribution and log2(x) transformed normalized H/L SILAC ratios were inverted, prior to averaging. Statistical significance was calculated via a one-sample, two-sided t test performed with truncation by a Benjamini Hochberg FDR (threshold value 0.05).

## Identification of cellular interactors using BioID

$4 \times 10^6$ WT MCF10A cells stably transduced with pCW57.tGFP, pCW57.tGFP.P2A.PTPRK.C1089S-BirA*-FLAG or pCW57.tGFP.P2A.PTPRK.1–785.BirA*-FLAG were seeded into 10 cm² dishes (three per condition). 24 hr after seeding, media was changed and doxycycline was added at 500 ng/ml for PTPRK.C1089S-BirA*-FLAG and tGFP, and 150 ng/ml for PTPRK.1–785-BirA*-FLAG. On the fourth day, doxycycline containing media was replaced and supplemented with 50 µM biotin (Sigma-Aldrich). After 24 hr, cells were lysed in 600 µl RIPA buffer (50 mM Tris–HCl pH 7.5, 150 mM NaCl, 1% (v/v) NP-40, 0.5% (w/v) sodium deoxycholate, 1 mM EDTA, 0.2% (w/v) SDS) and complete protease inhibitor cocktail (Roche). Cell lysates were then sonicated and clarified at 16500 x g for 10 min at 4°C. Equal amounts (3–4 mg) of lysate were transferred into 1.5 ml tubes which contained 50 µl of streptavidin agarose bead suspension (Thermo Fisher Scientific) that had previously been washed in RIPA buffer. Samples were then made up to 1 ml total volume in RIPA buffer and incubated at 4°C on a rotator overnight. Beads were pelleted at 14000 x g for 30 s at 4°C, supernatant removed and beads washed once with 1 ml 2% (w/v) SDS in PBS, followed by two washes with 1 ml 50 mM NaCl, 1% (v/v) NP-40, 50 mM Tris pH 7.5% and 0.2% (w/v) SDS including 8 min incubations on a rotator at RT. Proteins were eluted by incubation at 92°C in SDS sample buffer supplemented with 3 mM biotin for 10 min, prior to immunoblot analysis.

## In lysate dephosphorylation assay

All steps were performed on ice unless indicated. Each recombinant phosphatase domain was added to a total volume of 342 µl ice-cold 150 mM NaCl wash buffer to which 50 µl (200 µg) of freshly thawed pervanadate-treated cell lysate was added. Samples were mixed by gentle inversion and reactions were then incubated for 1.5 hr at 4°C on a rotator. 8 µl of 20% (w/v) SDS was then added (0.4% (w/v) final SDS concentration), samples were vortexed and incubated for 5–10 min. Samples were then diluted with 400 µl ice-cold 150 mM NaCl wash buffer to 0.2% (w/v) SDS final concentration and vortexed; prior to the addition of 5 µl of rabbit-anti-phospho-tyrosine antibody (Cell Signaling Technology). Samples were then incubated for 2–4 hr at 4°C on a rotator. 40 µl of washed protein G agarose bead suspension (Merck Millipore) was then added, prior to incubation overnight at 4°C on a rotator. Beads were pelleted at 15000 x g for 30 s at 4°C and washed five times in 1 ml of ice-cold 150 mM NaCl wash buffer. After the final wash, beads were transferred to RT and resuspended in SDS-PAGE sample buffer and incubated at 92°C for 10 min. Beads were pelleted at 15000 x g for 30 s and the supernatant transferred into a new microfuge tube. Samples were stored at −20°C prior to SDS-PAGE and immunoblot analysis.

## Protein structure presentation and homology modeling

All manipulations and homology modeling based on existing structures were performed using University of California San Francisco (UCSF) Chimera (*Pettersen et al., 2004*).

## Immunostaining MCF10A monolayers

$5 \times 10^5$ cells were seeded in 3 ml of complete growth medium on 18 mm x 18 mm,1.5 mm thick high-performance coverslips (Zeiss, UK). Cells were cultured for 6 days, with a media change on day

3, and then every day thereafter. On day 6, media was removed and cells fixed in 500 µl of metha-nol-free 4% (w/v) para-formaldehyde (PFA; Thermo Fisher Scientific) in PBS for 10 min at RT. Cover-slips were then rinsed with 5 × 500 µl of PBS, followed by permeabilization in 500 µl of 0.5% (v/v) triton X-100, 3% (w/v) BSA in PBS for 2 min at room temperature and blocking in 1 ml of 0.2% (v/v) triton X-100, 3% (w/v) BSA in PBS for 1 hr at RT. Coverslips were then incubated with primary anti-body (1:100 dilution) for 1–5 hr at RT. followed by five 500 µl 5 min washes with 0.2% (v/v) triton X-100, 3% (w/v) BSA in PBS. Coverslips were then incubated with species-specific fluorophore- con-jugated anti-IgG antibodies (1:250 dilution) containing Hoechst 33342 (1:2000; Thermo Fisher Scien-tific) with or without BODIPY 558/568 phalloidin (1:250; Thermo Fisher Scientific), for 45–60 min at RT in the dark. Coverslips were then rinsed twice with 500 µl 0.2% (v/v) triton X-100, 3% (w/v) BSA in PBS, followed by three 500 µl washes (5 min) with PBS. Coverslips were then mounted onto 1.0 mm thick slides using ProLong Gold antifade (Thermo Fisher Scientific). Slides were imaged using either a LSM880 confocal, LSM710 confocal, Elyra PS1 Super resolution or an AxioImager Z2 microscope (Zeiss).

## MCF10A spheroid cultures and immunostaining

MCF10As were cultured as spheroids following the previously described '3D on-top' method (*Lee et al., 2007*). 96-well plates were chilled for 30 min in the fridge before use. 15 µl Matrigel (Corning, Thermo Fisher Scientific), was spread evenly on the bottom of each well and allowed to set at 37°C for 20 min. $5 \times 10^3$ MCF10A cells per well were resuspended in MCF10A growth media and layered on top of the matrix and incubated for 20 min. 30 µl MCF10A growth media containing 10% (v/v) Matrigel was then added on top of the cells. Media was replaced with 30 µl complete growth media and 2% (v/v) Matrigel every 2–3 days for 7 days and then switched to EGF-free MCF10A growth media and 2% (v/v) Matrigel for 7 days. Acini were imaged at 10x magnification on day 14, using the EVOS FL Cell Imaging System (Thermo Fisher Scientific).

Spheroids were extracted from Matrigel as previously described (*Lee et al., 2007*). Media was aspirated and wells washed twice with PBS. Spheroids were extracted using 5 mM EDTA in PBS and gentle shaking for 30 min. Spheroids were then briefly centrifuged at 115 x g and the majority of supernatant aspirated. The remaining supernatant was used to resuspend the spheroids prior to transferring them onto a glass slide. Spheroids were fixed in 4% (v/v) paraformaldehyde (Thermo Fisher Scientific) for 20 min at RT and then permeabilized with 0.5% (v/v) Triton X-100 for 10 min at 4°C. The fixed spheroids were then washed three times in 100 mM glycine in PBS with 10 min per wash. Next, the spheroids were blocked in IF buffer (0.1% (w/v) BSA; 0.2% (v/v) Triton X-100; 0.05% (v/v) Tween-20) with 10% (v/v) normal serum block (BioLegend, USA) for 60 min at RT. Primary anti-body was incubated overnight at 4°C then washed three times with IF buffer. Secondary antibody was incubated for 45 min at RT. The spheroids were then washed once with IF buffer for 20 min, fol-lowed by two subsequent washes with PBS for 10 min each. They were then mounted with Prolong Gold Antifade Mounting medium (Thermo Fisher Scientific) and imaged using an LSM880 confocal or an AxioImager Z2 microscope (Zeiss).

## Quantification of confocal microscopy images

For immunostained cell monolayers, five random fields were imaged per condition and the results averaged. Image analysis was carried out using Fiji. The Pearson correlation coefficient for two images was determined using the Coloc2 plugin; whilst the fluorescence intensity of an image was analyzed using a custom macro: run('Auto Threshold', 'method = Default ignore_black white'); run ('Set Measurements...', 'integrated limit display redirect = None decimal = 3'); run('Measure').

For spheroids, aberrant spheroids were quantified using bright field images of 6 independent wells of WT and PTPRK KO MCF10A spheroids. three non-overlapping images from each well were manually counted for aberrant spheroids. Spheroid diameter was calculated using the circle mea-surement tool in Zen Pro (Zeiss). A circle was traced around individual spheroids in whole slide images for WT and PTPRK KO MCF10A spheroids using the Hoechst channel.

## BrdU incorporation ELISA

In a 96-well plate, $1 \times 10^4$ MCF10A cells per well were seeded in 90 µl of complete growth medium and cultured for 2 days. A final concentration of 10 µM bromodeoxyuridine (BrdU) was added to

each well and left to incorporate for 2 hr. A BrdU-based cell proliferation ELISA was then performed according to manufacturer's instructions (Roche, Germany). Absorbance was measured at 370 nm (reference wavelength 492 nm) using a 96-well plate reader (SpectraMax M5, Molecular Devices).

## Trans-epithelial electrical resistance (TEER)

$5 \times 10^5$ cells were seeded in 500 µl of complete growth medium onto the apical side of a 0.4 µm pore size Transwell filter (Corning) inserted into a 24-well companion plate containing (Corning) 500 µl of complete growth medium. Cells were cultured for 6–7 days to allow formation of a complete monolayer, with a media change day three and then every day thereafter. Growth medium was replaced 24 hr prior to TEER assessment; 5–10 readings were taken using a Millicell ERS-2 Volt/Ohm meter (Merck Millipore) and a mean was calculated. TEER value was calculated as follows: (Sample average TEER measurement ($\Omega$) – Blank average TEER measurement ($\Omega$)) x Trans-well surface area (0.3 cm$^2$).

## Fluorescein isothiocyanate (FITC)-dextran cell permeability assay

Cells were seeded on the apical side of transwell filters as described for TEER experiments. Growth medium was replaced 24 hr prior the addition of 250 kDa-FITC-dextran (3 mg/ml final concentration; Sigma-Aldrich) to the apical side of the insert; cells were then incubated for 24 hr. After 24 hr inserts were removed and the basal media was mixed by gentle pipetting. Per condition, $4 \times 100$ µl samples were transfer into a 96-well plate and the fluorescence intensity was measured using a 96-well plate fluorimeter (SpectraMax M5, Molecular Devices) at excitation 494 nm and emission at 515 nm.

## Accession codes

The mass spectrometry proteomics data have been deposited to the ProteomeXchange Consortium via the PRIDE partner repository (*Vizcaíno et al., 2016*) with the dataset identifier PXD013055.

# Acknowledgements

We thank FJ de Sauvage, A-C Gingras, V Pham, J Lill, D Ron and M Weekes for reagents and expertise; M Gratian and M Bowen for microscopy expertise; the CIMR flow cytometry core facility, in particular R Schulte and C Cossetti for their advice and support in cell sorting; G Griffiths, D Larrieu, S Munro, and A Schuldt for critical reading of the manuscript. This work was supported by a Sir Henry Dale Fellowship jointly funded by the Wellcome Trust and the Royal Society (109407) awarded to HJS and a Wellcome Trust (100140) grant awarded to CIMR. JED is funded by a Royal Society fellowship (100371). JRE is supported by a Wellcome Trust grant (086598). KAY is supported by a CRUK PhD studentship and IMH is supported by a CIMR PhD studentship.

# Additional information

## Competing interests

Wei-Ching Liang, Nadia Martinez-Martin, WeiYu Lin: Employed by Genentech and own Roche shares. The other authors declare that no competing interests exist.

## Funding

| Funder | Grant reference number | Author |
| --- | --- | --- |
| Wellcome and Royal Society | Sir Henry Dale Fellowship: 109407 | Gareth W Fearnley Hayley J Sharpe |
| Royal Society | 100371 | Janet E Deane |
| Cancer Research UK | PhD Studentship | Katherine A Young |
| Wellcome | 086598 | James R Edgar |

The funders had no role in study design, data collection and interpretation, or the decision to submit the work for publication.

## Author contributions
Gareth W Fearnley, Conceptualization, Validation, Investigation, Visualization, Methodology, Writing—review and editing; Katherine A Young, Investigation, Visualization, Methodology, Writing—review and editing; James R Edgar, Nadia Martinez-Martin, Investigation, Methodology; Robin Antrobus, Formal analysis, Visualization, Methodology, Project administration; Iain M Hay, Resources, Methodology, Writing—review and editing; Wei-Ching Liang, WeiYu Lin, Formal analysis, Methodology; Janet E Deane, Supervision, Methodology; Hayley J Sharpe, Conceptualization, Formal analysis, Supervision, Funding acquisition, Validation, Investigation, Visualization, Methodology, Writing—original draft, Project administration, Writing—review and editing

## Author ORCIDs
James R Edgar  https://orcid.org/0000-0001-7903-8199
Janet E Deane  https://orcid.org/0000-0002-4863-0330
Hayley J Sharpe  https://orcid.org/0000-0002-4723-298X

## Decision letter and Author response
Decision letter https://doi.org/10.7554/eLife.44597.035
Author response https://doi.org/10.7554/eLife.44597.036

# Additional files

## Supplementary files
• Transparent reporting form
DOI: https://doi.org/10.7554/eLife.44597.030

## Data availability
All data generated or analysed during this study are included in the manuscript and supporting files. Source data files have been provided for Figures 6, 7 and 8. Proteomics data have been submitted to PRIDE under accession code: PXD013055.

The following dataset was generated:

| Author(s) | Year | Dataset title | Dataset URL | Database and Identifier |
|---|---|---|---|---|
| Gareth W Fearnley, Iain M Hay, Robin Antrobus | 2019 | The homophilic receptor PTPRK selectively dephosphorylates multiple junctional regulators to promote cell-cell adhesion | https://www.ebi.ac.uk/pride/archive/projects/PXD013055 | PRIDE, PXD013055 |

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
