## [Decision Letter]

[Editors’ note: a previous version of this study was rejected before peer review, but the authors submitted for reconsideration. The first decision letter before peer review is shown below.]

Thank you for choosing to send your work entitled "Systematic substrate identification indicates a role for the receptor tyrosine phosphatase PTPRK in epithelial cell-cell adhesion" for consideration at *eLife*. Your initial submission has been assessed by a Senior Editor in consultation with a member of the Board of Reviewing Editors. Although the work is of interest, we are not convinced that the findings presented have the potential significance that we require for publication in *eLife*. However, we would encourage you to resubmit if you can provide an additional piece of data, as described below.

The paper presents a detailed study of a receptor type protein tyrosine phosphatase, PTPRK, which was suspected to form homophilic interactions and respond to cell contact. PTPRK gene fusions are found in some cancers, and expression is induced by TGFb. The authors used quantitative proteomics with a "substrate trapping" mutant, expressed in bacteria, to pull out and identify potential substrates. Specificity was confirmed by Westerns, using the related PTP, PTPRM, as a specificity control. They then made PTPRK KO MCF10A cells and identified pY sites undergoing increased phosphorylation. There was an increase in several cell-cell junction proteins, including ones identified as binding to the substrate trapping mutant. BioID confirmed interactions between PTPRK and the proposed substrates in confluent MCF10A cells. Substrates were then tested for dephosphorylation by PTPRK in vitro. MCF10A cell lysates were incubated with recombinant PTPRK. p120Ctn and Afadin (AF6) were dephosphorylated and the sites identified. Three other substrates were identified with high confidence and several others as good candidates. They demonstrate the importance of the pseudophosphatase domain, as well as the phosphatase domain, in substrate recognition. Finally, PTPRK KO alters junction status in MCF10A cells, and alters growth and 3D acini formation. The data showing discrete colocalization of PTPRK and junctional proteins are also nice.

What sets this paper apart is that many different proteomics approaches were used to identify proteins that appear to be substrates in vivo. The use of MCF10A knockouts and reconstituting with wildtype PTPRK or the phosphatase dead mutant is a real plus. However, what is missing is direct evidence of the importance of any of the identified PTPRK sites in cell-cell junction proteins in the integrity of cell-cell junctions, or whether D1 PTP activity is required to reverse the observed MCF10A cell phenotypes. Reviewers are bound to ask and it would not be productive send it for review only to have it rejected because the significance of the phosphatase activity isn't clear. We would not expect the relevance of individual substrates or pY sites to be established, but it is important to test whether the PTP activity is required.

Less major points include whether the in vitro interactions with the WT PTPRK ICD or the substrate trapping mutant D1 domains were in fact dependent on pTyr residues, i.e. they did not treat the lysate with recombinant PTP prior to doing the pulldowns. A significant number of proteins bound selectively to the WT PTPRK ICD vs beads, but since the D1 catalytic domain was active, presumably these interactions were not pTyr dependent. This raises the general issue of the extent to which PTPRK interacting proteins associate directly via a target pTyr as opposed to secondary interactions, for instance those with the inactive D2 domain. In this regard, their model that D2 might recruit targets for D1 does not make much sense, unless the protein has two pTyr residues, since a pTyr bound to D2 would be protected from dephosphorylation by D1.

Another issue that is not discussed is whether there is a primary sequence preference for sites dephosphorylated by D1 – they have a number of identified sites but there is no sequence comparison.

[Editors’ note: what now follows is the decision letter after the authors submitted for in depth peer review.]

Thank you for submitting your article "The homophilic receptor PTPRK selectively dephosphorylates multiple junctional regulators to promote cell-cell adhesion" for consideration by *eLife*. Your article has been reviewed by three peer reviewers, including Tony Hunter as the Reviewing Editor and Reviewer #1, and the evaluation has been overseen by Jonathan Cooper as the Senior Editor. The following individuals involved in review of your submission have agreed to reveal their identity: Αlpha Yap (Reviewer #2); Michel Tremblay (Reviewer #3).

The reviewers have discussed the reviews with one another and the Reviewing Editor has drafted this decision to help you prepare a revised submission.

As you will see, two out of the three reviewers would have liked to see evidence that dephosphorylation of one or more of the PTPRK target proteins you identified is in fact important for regulating junctional integrity, but after discussion, we have decided that your new data showing that PTPRK activity is important for establishment of cell-cell junctions will suffice for this first paper. No further experiments are required, but please address the other points that the reviewers raised concerning the revised version of your paper. Thank you for submitting your paper to *eLife*.

Reviewer #1:

Given the reported connections between loss of function mutations in the PTPRK receptor phosphatase and cancer progression, which implicate it as a tumor suppressor, the authors' efforts to define specific substrates for PTPRK are important. They have used a wide variety of different approaches to achieve this goal, including recombinant substrate trapping protein pull down AP-MS proteomics combined with BirA tagging BioID detection of proteins in the vicinity of PTPRK in cells, and pTyr proteomics with PTPRK knockout MCF10A cells, reconstituted with WT or a catalytically-dead PTPRK mutant, to identify pTyr sites that changed when PTPRK activity was increased or decreased. These studies led to the identification of Afadin, PARD3, p120 catenin and PKP3 and PKP4, proteins that are all functionally linked to cell-cell junctions, as candidate specific substrates for PTPRK. By in vitro dephosphorylation of cell lysate proteins with recombinant PTPRK-ICD, using recombinant PTPRM-ICD as a specificity control, they identified two pTyr sites in p120 catenin that could be directly dephosphorylated by PTPRK. Finally, using the MCF10A PTPRK knockout cells, they showed that these cells exhibit defects in E-cadherin junctional contacts, and that re-expression of WT, but not catalytically-dead PTPRK, was able to partially restore the presence of E-cadherin at cell-cell junctions.

The identification of a small set of specific pTyr substrates for PTPRK is an advance in our understanding of this receptor PTP. The known junctional functions of the specific substrates clearly points towards a role for PTPRK in controlling cell-cell junctions, and the confirmatory evidence that PTPRK D1 catalytic activity is required for formation of proper cell-cell junctions and establishment of transepithelial electrical resistance is reasonable, although the rescue observed upon re-expression of PTPRK in the knockout cells was only partial. However, although in the resubmitted version, which now contains a massive amount of data, the authors addressed many of the concerns raised about the original version of the paper, they have fallen short of establishing that PTPRK-mediated dephosphorylaion of any of one the identified substrates is important for cell-cell junctions. It would strengthen the paper if they could provide evidence for this by expressing a mutant form of one (or more) of their substrates in which the PTPRK pTyr sites are mutated.

1) All the proteomic experiments were carried out with exogenously expressed PTPRK constructs. Did the authors carry also out a conventional IP/MS analysis of proteins associated with endogenous PTPRK using the anti-PTPRK mAb they have generated (an alternative would be to knock a tag into the PTPRK locus) to compare with the substrate trapping lysate pull down data and confirm interaction with the identified targets? Admittedly, this will be more challenging, but the endogenous PTPRK will be physically adjacent to likely interactors at epithelial junctions, and this could promote interactions not so readily detectable by pull downs from cell lysates where the proteins will be diluted and the contents of all the different cellular compartments will be mixed. In part their BioID data are informative in establishing that some of the candidate substrates are in the vicinity of PTPRK in the cell, but they did not use BioID to look for additional substrates for PTPRK.

2) Did the authors compare pTyr proteomes of confluent and sparse WT and knockout MCF10A cells to determine if cell-cell contact triggers PTPRK-dependent dephosphorylation events?

3) The evidence that some of the substrate-PTPRK interactions are independent of catalytic function implies, as the authors suggest, that PTPRK also has a scaffolding unction. However, they provide no insights into how these scaffolding interactions occur (i.e. what domains are involved?), and so we do not learn how important such interactions are for dephosphorylation of these substrates in the cell (or in vitro).

Reviewer #2:

This is an informative and rather heroic paper that identifies potential substrates for the membrane-bound receptor tyrosine phosphatase K (RPTPK). The authors identify and reasonably-validate a number of substrates and illustrate the complexities of tackling this problem (phosphatase-independent interactions etc). Many of these candidates are interesting proteins found at cell-cell junctions (as might be predicted from the junctional localization of RPTPK) and they include proteins, such as afadin, for which the functional consequences of dynamic phosphorylation are not well characterized. Overall, I think that the manuscript already presents a valuable resource for the community that will be the basis for future research.

Given the wealth of data that is already here, I hesitate to ask for more. But a limitation is that the authors don't have evidence for functional significance of any of the dephosphorylation events that they document. They show that RPTPK supports aspects of junctional integrity (especially barrier function) in a PTP-dependent fashion, but the impact of the manuscript would be enhanced if they could test the principle that one/some of their substrates contributes to this phenotype. The obvious candidate here is p120-catenin which they show to have increased phosphorylation on Y228 and Y904 in RPTPK KO cells. Is this hyperphosphorylation contributing to the epithelial phenotype that they document? One way to test this hypothesis would be to express Y228A/Y904A p120 in their PRPTK KO cells. (Most elegantly on a p120 RNAi background – for which there are good siRNAs available; but overexpression may be effective.)

Reviewer #3:

General comments on the manuscript:

The general interest of this manuscript to characterize specific substrates, is very important on a larger scheme of understanding RPTP (R2Bs) mechanism of action. This search for PTPRK substrates has also significant importance considering the "tumor suppressive" activities associated with PTPRK function in many cancers. It is in part answering the need to identify PTPRK substrates as potential mechanism of oncogenic development. The premises and general findings of the manuscript are of great interest in the PTP field and beyond.

Looking at the results presented, the experimental methods are well described and for the most part standard. The workflow drawings are simple and useful. Statistical analysis using anova is also appropriate for the study.

In conclusion, the manuscript is an excellent example of a comprehensive biochemical analysis of a receptor PTP and a search for physiological substrates. On this, the manuscript has interest for the readerships. Yet without a solid validation of the putative substrates into a physiological cell or animal model of diseases, the manuscript comes out short on the biological value of these findings.

Considering the rebuttal letter, four specific questions were answered by authors in this resubmission.

- Comments on answer to Question 1.

These are indeed valuable experiments. Both are difficult to generate. The first one is complicated since it is not known which substrate phospho-tyrosine site(s) would be crucial for cell -adhesion. As for the PTPRK rescue of the PTP knock-out cells, authors have already stated that the transfected PTPRK cDNA is not expressed and likely to be at various levels.

In both approaches, authors could have used various experimental design such as introducing into a safe harbor site (i.e.: AAV1,.…) the dox dependent PTPRK expression vector into the knockout cell line, and test cell-cell phenotype rescue.

A similar approach may have perhaps been attempted with the CRISPR KO of the specific substrate(s), follow by their rescue through re-expressing their cDNAs with mutated p-tyr in the same safe harbor site with dox inducible re-expression. This would then be validating the phospho-sites specific effect on cell-cell interaction. It remains that with several identified and predicted substrates' sites reported in this manuscript, this task although desirable would be major undertaking to perform.

- Comments on answer to Question 2.

On this issue, data presented are supporting that some substrates (i.e. afidin,) are indeed binding to the D2 domain in a phosphorylation independent manner and then are likely being dephosphorylated. This is a common feature of many PTP substrate recognition where other protein-protein domains anchor them to a phosphatase and then facilitate their dephosphorylation. The manuscript accurately makes a point that this is also occurring with some substrate of RPTPs.

Yet they have not identified the specific binding sites of afidin to the D2 domain. Several receptor PTPs have, like RTK, C terminal ends with additional features such as phosphorylation sites. This does not seem to be the case with the very short C-tail of the PTPRK.

- Comments on answer to Question 3

The authors' answer is unsatisfying. D2 domain for many RPTPs were tested as per their potential to trap P-Tyr proteins and there are no cases where these occur. D2 domains were associated to important regulatory functions of RPTPs such as stabilizing substrate interactions, mediating protein-protein interactions, protection against Cysteine-Oxydation and facilitating RPTP dimerization causing inhibitory phosphatase activities. A stated above, in some cases, such as PTPRE and PTPRA, the C-terminus of the RPTP provide phosphorylation dependent docking outside the D2 domain. The only solution on this issue would be to map the interactions of afidin to the D2 domain. Better of course, a structure of the PTPRK D2- afidin interaction would be ideal in understanding this mechanism. This is out of scope for this manuscript.

- Comments on answer to Question 4

The answer by authors to generate a consensus sequence is of interest particularly with the additional list of putative substrates that was generated. Yet, without validation of some of them in the KO cells/mouse… this remains speculation.

As for the statement that this effort is the first unbiased proteomic approach to identify substrates for classical PTPs, this is inaccurate as among many reports, those of Mertins et al. Mol Cell Proteomics. 2008, Lee and Bennett Methods Mol Biol. 2015, Tang et al. J. Am. Chem. Soc., 2018 are just excellent examples of such approaches.

---

## [Author Response]

[Editors’ note: the author responses to the initial decision follow.]

1) What sets this paper apart is that many different proteomics approaches were used to identify proteins that appear to be substrates in vivo. The use of MCF10A knockouts and reconstituting with wildtype PTPRK or the phosphatase dead mutant is a real plus. However, what is missing is direct evidence of the importance of any of the identified PTPRK sites in cell-cell junction proteins in the integrity of cell-cell junctions, or whether D1 PTP activity is required to reverse the observed MCF10A cell phenotypes. Reviewers are bound to ask and it would not be productive send it for review only to have it rejected because the significance of the phosphatase activity isn't clear. We would not expect the relevance of individual substrates or pY sites to be established, but it is important to test whether the PTP activity is required.

We appreciate that linking alterations in signaling to the observed phenotypes will raise the impact of this work, and as you point out, would be an obvious reviewer comment. However, these experiments are particularly challenging because of a lack of expression level uniformity, which one would anticipate to be particularly important for cell-cell adhesion proteins. Nevertheless, we now include experiments that show that PTPRK partially rescues TEER and colocalization of E-Cadherin and F-actin in MCF10A PTPRK KO cells, in a phosphatase-dependent manner. Unfortunately, we could not detect induction of PTPRK in MCF10A spheroids after 14 days, which would be a key control required to interpret any changes. We conclude that phosphatase activity is an important part of the PTPRK cell adhesion phenotype, however, we also believe there is a key scaffolding function for this receptor, which is highlighted by experiments outlined below.

2) Less major points include whether the in vitro interactions with the WT PTPRK ICD or the substrate trapping mutant D1 domains were in fact dependent on pTyr residues, i.e. they did not treat the lysate with recombinant PTP prior to doing the pulldowns. A significant number of proteins bound selectively to the WT PTPRK ICD vs beads, but since the D1 catalytic domain was active, presumably these interactions were not pTyr dependent.

This is a good point and we too had assumed initially that interactions would depend on phosphotyrosine. The two traps used do have different properties, one allows the formation of a phosphocysteine intermediate (DA), the other relies on the increased affinity of enzyme for its substrate over product (CS). We agree with your analysis that binding of proteins to the WT ICD suggests phosphotyrosine is not critical for the recruitment of most substrates. There are examples in the literature, now included in our Discussion, where binding of substrates to other active, wildtype classical PTPs is observed. In the case of MAP4K4, which was trapped on the DA mutant, we have now demonstrated that this interaction can be completed by vanadate, suggesting it is trapped by phosphotyrosine. In addition, PARD3, PKP4 and p120^Cat^ all show reduced binding to the CS trap after pulldowns from calf intestinal phosphatasetreated pervanadate lysates, again suggesting a role for phosphorylation in trapping. Surprisingly, however, we show that most substrates still bind efficiently to PTPRK domains in the absence of phosphorylation. In addition, our data support a key role for the pseudophosphatase domain in recognizing several substrates, which leads to the next issue highlighted.

3) This raises the general issue of the extent to which PTPRK interacting proteins associate directly via a target pTyr as opposed to secondary interactions, for instance those with the inactive D2 domain. In this regard, their model that D2 might recruit targets for D1 does not make much sense, unless the protein has two pTyr residues, since a pTyr bound to D2 would be protected from dephosphorylation by D1.

We had also initially anticipated that the D2 domain would function as a pTyr trap. This is because key residues in catalytic motifs of the D2 domain (HCS, WPD and Q loop) deviate from D1 sequences and resemble canonical substrate trapping mutants. Previously, the D2 domain of PTPRF (LAR) was reactivated by reverting particular sequences to those found in D1 domains. In fact, we tried exactly this with PTPRK but did not see any effect on D2 activity or binding of key substrates (data now included). This matches with recent results for PTPRE, where the D2 domain could not be reactivated by similar mutations (Lountos et al. PMID:30289412). We also noted that for existing D2 domain crystal structures, the catalytic cysteine is largely occluded (e.g. CD45; Figure 5—figure supplement 2). Unfortunately, there is not a PTPRK-family phosphatase D2 domain structure available, however, we have included a homology model that indicates significant changes in surface charge compared to the D1 such that it would be unlikely to accommodate a phosphate group. Finally, orthovanadate or lysate dephosphorylation was ineffective at depleting Afadin from the D2 domain. We think substrate recognition, in a phosphorylation-independent manner, will be an important area for future research and indicates a putative scaffolding role for these receptors, comprising dephosphorylated proteins.

4) Another issue that is not discussed is whether there is a primary sequence preference for sites dephosphorylated by D1 – they have a number of identified sites but there is no sequence comparison.

We generated a consensus sequence using our high confidence and “likely” substrates and now present a weblogo (Figure 3—figure supplement 1). The predominant trend we observe is the absence of basic residues in the pTyr-1 and pTyr+1 positions, which is consistent with the highly positively charged nature of the PTPRK active site excluding interactions with such amino acids. Similar attempts to map a substrate consensus sequence for PTP1B also show a preference for acidic or negatively charged residues N-terminal of the pTyr (Li et al. PMID: 23674824). Thus, there are probably permissive substrate primary sequences, however, given the exquisite selectivity of PTPRK it seems unlikely that primary sequence is the major specificity determinant. As an aside, we revisited work by Barr et al. (PMID: 19167335) that showed PTPRK does not dephosphorylate an EGFR-pY1068 peptide, supporting our work at the protein level where we do not observe dephosphorylation of this site. This is a particularly important negative result as EGFR-pY1068 is one of the most highly cited PTPRK substrates (Xu et al. PMID: 16263724).

In terms of significance, this is the first unbiased proteomic approach to identify substrates for a classical PTP. Our data suggest that the domains beyond the active enzyme play a key role in substrate recognition, which has also been suggested for some individual PTP substrates (e.g. Timms et al. PMID: 9632768). Our work is of high quality and provides a key step forward for the phosphatase field, by challenging general assumptions about RPTPs, for example, as thresholders of receptor tyrosine kinase signaling (e.g. Lee and Bennett PMID: 25319894) and raises the possibility that they function as core signaling scaffolds, similar to RTKs. Moreover, we have shown that PTPRK is a convincing and exciting new player in cell-cell adhesion and epithelial to mesenchymal transition. It has the potential to function as a sensor of cell contact, and associates with several known mechanotransduction molecules such as MAP4K4 and RAPGEF6. Thus, two major new questions arise from our work: How (and why) do RPTP pseudophosphatase domains recognize substrates? Is PTPRK an upstream component of mechanosensing and/or mechanotransduction pathways?

[Editors' note: the author responses to in depth peer review follow]

Reviewer #1:[…] The identification of a small set of specific pTyr substrates for PTPRK is an advance in our understanding of this receptor PTP. The known junctional functions of the specific substrates clearly points towards a role for PTPRK in controlling cell-cell junctions, and the confirmatory evidence that PTPRK D1 catalytic activity is required for formation of proper cell-cell junctions and establishment of transepithelial electrical resistance is reasonable, although the rescue observed upon re-expression of PTPRK in the knockout cells was only partial. However, although in the resubmitted version, which now contains a massive amount of data, the authors addressed many of the concerns raised about the original version of the paper, they have fallen short of establishing that PTPRK-mediated dephosphorylaion of any of one the identified substrates is important for cell-cell junctions. It would strengthen the paper if they could provide evidence for this by expressing a mutant form of one (or more) of their substrates in which the PTPRK pTyr sites are mutated.

Although we show that PTPRK phosphatase activity is required to rescue signalling and cell adhesion phenotypes in KO cells, we agree that additional evidence showing that substrate hyperphosphorylation directly influences junctional integrity would substantially strengthen the paper. However, this would be a significant undertaking, out of scope for this initial study (as agreed by the editors). We believe this would be challenging for several reasons. There are many phosphosites requiring characterisation (> 15 upregulated in PTPRK KO cells), which may, in combination, contribute to the observed phenotypes. Therefore, rescues with phosphorylation-deficient mutant substrates might be masked by the effects of other hyperphosphorylated substrates. Moreover, some substrates have multiple phosphosites to functionally analyse, such as p120^Cat^ where 4 phosphosites are altered in PTPRK KO cells. The ideal experiment would be to overexpress phosphomimetic mutants to see if the KO phenotypes are recapitulated on a WT background, but such mutants are unreliable for phosphotyrosine.

In an attempt to further strengthen the notion that substrate hyperphosphorylation is contributing to the PTPRK KO junctional phenotype, we have added data showing that p120^Cat^ (hyperphosphorylated under these conditions) exhibits reduced junctional intensity by immunofluorescence in PTPRK KO vs WT MCF10A. In addition, there are several papers that have investigated the functions of some of the dysregulated sites in other settings, including p120^Cat^-Y228 and Y904 and Afadin Y1230, therefore we have expanded our Discussion to include these references.

1) All the proteomic experiments were carried out with exogenously expressed PTPRK constructs. Did the authors carry also out a conventional IP/MS analysis of proteins associated with endogenous PTPRK using the anti-PTPRK mAb they have generated (an alternative would be to knock a tag into the PTPRK locus) to compare with the substrate trapping lysate pull down data and confirm interaction with the identified targets? Admittedly, this will be more challenging, but the endogenous PTPRK will be physically adjacent to likely interactors at epithelial junctions, and this could promote interactions not so readily detectable by pull downs from cell lysates where the proteins will be diluted and the contents of all the different cellular compartments will be mixed. In part their BioID data are informative in establishing that some of the candidate substrates are in the vicinity of PTPRK in the cell, but they did not use BioID to look for additional substrates for PTPRK.

The PTP substrate trapping approach requires hyperphosphorylated substrates, which is readily achieved using pervanadate. IP/MS and BioID/MS approaches are less suitable than pull downs, as pervanadate would interfere with substrate binding for a D>A mutant in a cellular context. Indeed, phosphorylation of some putative PTPRK substrates was critical for successful enrichment using substrate-trapping mutants (e.g. MAP4K4 binding to the D1057A substrate-trapping mutant). However, we now know that phosphorylation-independent interactions between PTPRK and its substrates occur meaning IP/MS and BioID would identify PTP substrates. The Gingras lab has already performed a PTPRK BioID screen (St Denis et al., 2016) and reassuringly several of our PTPRK interactors (and substrates) overlap including PKP2, KIF14, DLG5, DNAJA3, PKP4, PLEKHA5 and PTPN14, despite the use of different cell lines. Recently, IP/MS was also used to identify PTP interactors (Kumar P et al., 2017) in HEK293T cells. This is a less favourable approach for the receptor PTPs due to compromises between solubility and preserving interactions. Indeed, Kumar et al. did not enrich for junctional interactors for PTPRM by IP/MS from HEK293T cells, however, PTPRU interacted with some junctional proteins including Afadin. Overall, the number of interactors is notably fewer for PTPRM and PTPRU than we find for PTPRK. Our main aim was to identify high-confidence substrates for PTPRK, however, we agree that a combination of approaches, in different cell lines, would identify additional substrate candidates.

2) Did the authors compare pTyr proteomes of confluent and sparse WT and knockout MCF10A cells to determine if cell-cell contact triggers PTPRK-dependent dephosphorylation events?

We did not do this experiment; however, it is an interesting suggestion. Wild type MCF10A cells experience cell-cell contact when subconfluent, tending to grow in tightly associated ‘islands’ that collectively migrate to form a confluent monolayer. PTPRK localises to filopodial contacts between individual cells. In line with the presence of PTPRK at early contacts, we note that PTPRK KO cells do not coalesce to the same extent as wildtype cells (see Author response image 1), suggesting an early adhesion defect. It is therefore likely that signaling differences are occurring even at this early stage and would be worth analysing, with the caveat that contact-mediated kinase signaling (e.g. Eph receptors) might be compromised at low confluence.

3) The evidence that some of the substrate-PTPRK interactions are independent of catalytic function implies, as the authors suggest, that PTPRK also has a scaffolding unction. However, they provide no insights into how these scaffolding interactions occur (i.e. what domains are involved?), and so we do not learn how important such interactions are for dephosphorylation of these substrates in the cell (or in vitro).

We do provide evidence that the D2 domains of PTPRK and PTPRM can determine interaction specificity (NUFIP2, Afadin), however, we concede that the domains of PTPRK substrates that determine their interactions have not been mapped. These will be important future experiments to understand the full range of signalling that can be mediated by these receptors through their substrates and interaction partners.

Reviewer #2:[…] Given the wealth of data that is already here, I hesitate to ask for more. But a limitation is that the authors don't have evidence for functional significance of any of the dephosphorylation events that they document. They show that RPTPK supports aspects of junctional integrity (especially barrier function) in a PTP-dependent fashion, but the impact of the manuscript would be enhanced if they could test the principle that one/some of their substrates contributes to this phenotype. The obvious candidate here is p120-catenin which they show to have increased phosphorylation on Y228 and Y904 in RPTPK KO cells. Is this hyperphosphorylation contributing to the epithelial phenotype that they document? One way to test this hypothesis would be to express Y228A/Y904A p120 in their PRPTK KO cells. (Most elegantly on a p120 RNAi background – for which there are good siRNAs available; but overexpression may be effective.)

Please see comments addressing a similar point from reviewer #1. This is a great suggestion, however, one of the challenges we could anticipate is achieving similar levels of expression in neighbouring confluent cells to achieve a rescue of junctional integrity. Constructs for p120^Cat^ with phosphorylation-deficient mutants at Y174, Y228, Y904 and Y865 would have to be stably integrated via lentiviral transduction or introduced via a safe harbour site (as suggested by reviewer #3) into PTPRK KO MCF10A cells. This experiment has been done in part demonstrating that phosphorylation deficient mutants affecting tyrosines 112, 228, 257, 280, 291, 296, 302 could rescue p120^Cat^ null adhesion phenotypes (Mariner et al., 2004). This is at least consistent with the idea that dephosphorylated p120^Cat^ promotes adhesion. It might be simpler to test the impact of Y1230F on Afadin. However, Afadin knockdown has a dramatic impact on cell-cell adhesion meaning the mutant form would have to rescue this first, which might make interpreting PTPRK KO junctional phenotypes challenging. Furthermore, Afadin is a large protein (>1800 AAs) presenting additional challenges with cell line generation.

Reviewer #3:[…] Considering the rebuttal letter, four specific questions were answered by authors in this resubmission.- Comments on answer to “However, what is missing is direct evidence of the importance of any of the identified PTPRK sites in cell-cell junction proteins in the integrity of cell-cell junctions, or whether D1 PTP activity is required to reverse the observed MCF10A cell phenotypes.”.These are indeed valuable experiments. Both are difficult to generate. The first one is complicated since it is not known which substrate phospho-tyrosine site(s) would be crucial for cell -adhesion. As for the PTPRK rescue of the PTP knock-out cells, authors have already stated that the transfected PTPRK cDNA is not expressed and likely to be at various levels.In both approaches, authors could have used various experimental design such as introducing into a safe harbor site (i.e.: AAV1,.…) the dox dependent PTPRK expression vector into the knockout cell line, and test cell-cell phenotype rescue.A similar approach may have perhaps been attempted with the CRISPR KO of the specific substrate(s), follow by their rescue through re-expressing their cDNAs with mutated p-tyr in the same safe harbor site with dox inducible re-expression. This would then be validating the phospho-sites specific effect on cell-cell interaction. It remains that with several identified and predicted substrates' sites reported in this manuscript, this task although desirable would be major undertaking to perform.

For rescue experiments, although our lentiviral stable cell lines were sorted by flow cytometry based on their expression levels, there was still intercellular variability that could have influenced the degree of rescue achievable. A safe harbour might have been a good alternative, however, the size of these cDNAs makes them challenging to engineer into cell lines. We have also found that the *PTPRK* cDNA is susceptible to recombination. We agree that attempting to understand the contributions of the individual substrate sites would be informative, but also a major undertaking beyond the scope of this manuscript, as described in the comments to reviewers #1 and #2.

- Comments on answer to “whether the in vitro interactions with the WT PTPRK ICD or the substrate trapping mutant D1 domains were in fact dependent on pTyr residues, i.e. they did not treat the lysate with recombinant PTP prior to doing the pulldowns. A significant number of proteins bound selectively to the WT PTPRK ICD vs beads, but since the D1 catalytic domain was active, presumably these interactions were not pTyr dependent.”.On this issue, data presented are supporting that some substrates (i.e. afidin,) are indeed binding to the D2 domain in a phosphorylation independent manner and then are likely being dephosphorylated. This is a common feature of many PTP substrate recognition where other protein-protein domains anchor them to a phosphatase and then facilitate their dephosphorylation. The manuscript accurately makes a point that this is also occurring with some substrate of RPTPs.Yet they have not identified the specific binding sites of afidin to the D2 domain. Several receptor PTPs have, like RTK, C terminal ends with additional features such as phosphorylation sites. This does not seem to be the case with the very short C-tail of the PTPRK.

As the reviewer points out, there are limited additional sequence features beyond the PTPRK D2 domain, suggesting that substrates, such as Afadin, may be binding to surfaces on the domain itself. An interesting future question is why these receptors possess pseudo-phosphatase domains rather than other protein-protein interaction domains, such as PDZ-binding domains present in, for example, Eph receptors.

- Comments on answer to “In this regard, their model that D2 might recruit targets for D1 does not make much sense, unless the protein has two pTyr residues, since a pTyr bound to D2 would be protected from dephosphorylation by D1.”The authors' answer is unsatisfying. D2 domain for many RPTPs were tested as per their potential to trap P-Tyr proteins and there are no cases where these occur. D2 domains were associated to important regulatory functions of RPTPs such as stabilizing substrate interactions, mediating protein-protein interactions, protection against Cysteine-Oxydation and facilitating RPTP dimerization causing inhibitory phosphatase activities. A stated above, in some cases, such as PTPRE and PTPRA, the C-terminus of the RPTP provide phosphorylation dependent docking outside the D2 domain. The only solution on this issue would be to map the interactions of afidin to the D2 domain. Better of course, a structure of the PTPRK D2- afidin interaction would be ideal in understanding this mechanism. This is out of scope for this manuscript.

The R2B family D2 domain has not yet been characterized structurally, we therefore did not want to make any assumptions about the nature of the interactions. A previous study showed that the PTPRF/LAR D2 domain could be reactivated by mutations in two motifs (Nam et al., 1999) suggesting that its catalytic cysteine is accessible to phosphotyrosine. Indeed, studies on DLAR suggest its D2 domain binds phosphopeptides (Madan et al., 2011). The PTPRK D2 domain possesses a WPD>WAS mutation in its WPD-Loop, resembling a D>A substrate trapping mutant. However, as pointed out, this is not the mechanism for D2 domain recognition of substrates, which remains an open question. A co-crystal structure is an important future goal, first requiring domain mapping experiments, for example, with Afadin as suggested by reviewer #1.

- Comments on answer to “Another issue that is not discussed is whether there is a primary sequence preference for sites dephosphorylated by D1 – they have a number of identified sites but there is no sequence comparison.”The answer by authors to generate a consensus sequence is of interest particularly with the additional list of putative substrates that was generated. Yet, without validation of some of them in the KO cells/mouse… this remains speculation.

Although the reviewer suggests that the outcome of this analysis is speculative, it agrees with previous studies that show limited specificity at the sequence level for several PTPs (e.g. Barr et al., 2009). We agree that more validated PTP substrates are required to establish this.

As for the statement that this effort is the first unbiased proteomic approach to identify substrates for classical PTPs, this is inaccurate as among many reports, those of Mertins et al. Mol Cell Proteomics. 2008, Lee and Bennett Methods Mol Biol. 2015, Tang et al. J. Am. Chem. Soc., 2018 are just excellent examples of such approaches.

This is an incorrect statement on our part, and was not intended to diminish the efforts of numerous previous studies. However, we do believe this is the first of such approaches to identify and validate substrates for the R2B RPTP subfamily. Thank you for allowing us to clarify.